# Climatic windows for human migration out of Africa in the past 300,000 years

Robert M. Beyer[1,2✉], Mario Krapp[1,3], Anders Eriksson[4,5] & Andrea Manica[1✉]

Whilst an African origin of modern humans is well established, the timings and routes of their expansions into Eurasia are the subject of heated debate, due to the scarcity of fossils and the lack of suitably old ancient DNA. Here, we use high-resolution palaeoclimate reconstructions to estimate how difficult it would have been for humans in terms of rainfall availability to leave the African continent in the past 300k years. We then combine these results with an anthropologically and ecologically motivated estimate of the minimum level of rainfall required by hunter-gatherers to survive, allowing us to reconstruct when, and along which geographic paths, expansions out of Africa would have been climatically feasible. The estimated timings and routes of potential contact with Eurasia are compatible with archaeological and genetic evidence of human expansions out of Africa, highlighting the key role of palaeoclimate variability for modern human dispersals.

[1] Department of Zoology, University of Cambridge, Cambridge, UK. [2] Potsdam Institute for Climate Impact Research (PIK), Member of the Leibniz Association, Potsdam, Germany. [3] GNS Science, Lower Hutt, New Zealand. [4] cGEM, cGEM, Institute of Genomics, University of Tartu, Tartu, Estonia. [5] Department of Medical and Molecular Genetics, King's College London, London, UK. ✉email: robert.beyer@pik-potsdam.de; am315@cam.ac.uk

Analysis of fossil and genetic evidence provides strong support for an African origin of *Homo sapiens*[1], yet the timing of human expansions out of Africa has been the focus of recent debate[2]. The dating of most out-of-Africa fossils, and of the split between Eurasian and African populations based on mitochondrial and whole-genome data[3,4] point to a major exit ~65k years ago. However, archaeological findings in Saudi Arabia dated to at least 85k years ago[5], in Israel to at least 100k years ago[6] and possibly as old as 194k years ago[7] (challenged[8] and defended[9]), and in Greece to 210k years ago[10] hint at previous excursions from Africa, which might have reached China at least 80k, possibly 120k years ago[11] (challenged[12] and defended[13]). These, or other previous waves, might have also left a small genetic contribution (~1%) found in modern inhabitants of Papua New Guinea[14]. Further evidence for a possibly earlier exit comes from traces of geneflow from *Homo sapiens* into Neanderthal, genetically dated to before 130k years ago[15,16], and which could be as old as 250k years ago[17]. While more work is needed to confirm hints of Middle Pleistocene humans in Eurasia, current evidence strongly suggests that *Homo sapiens* had repeated periods when they were able to leave Africa, although the number, timings, routes, and fates of such early waves is unclear.

Palaeoclimate reconstructions can provide insights into possible windows out of Africa. Most studies have discussed qualitatively possible scenarios, using reconstructions of climatic conditions in Northern Africa based either on empirical records[18–23], which are too sparse to provide a spatially complete picture of climatically viable migration paths into Eurasia, or on model-based data from a few time slices[24,25]. Quantitative attempts to define possible windows out of Africa have fitted demographic rules in human dispersal models to match either the archaeological record[26] or genetic data[27]. While it is possible to find such rules, it is unclear how biologically realistic they are (e.g. in ref. [26], it is assumed that the thermal niche of humans changed by 50 °C and that coastal migration speed increased by almost 6000% over a period of 125k years). Moreover, the archaeological record is very sparse, especially for the early periods, and genetic data only reflect exits whose descendants have been sampled. Here, we take a different approach to identify suitable periods when *Homo sapiens* might have left Africa. First, we use high-resolution palaeoclimate simulations of the last 300k years to estimate the tolerance to low precipitation and aridity that would have been required for humans to successfully exit Africa at a given time. In a second step, we combine these data with estimates of the actual climatic tolerances of hunter gatherers based on anthropological and ecological data, allowing us to reconstruct the timings of climatic windows out of Africa. We then examine how compatible our inferred periods of connectivity are with the available empirical archaeological and genetic evidence of out-of-Africa expansions.

Our analysis reveals that exit routes and timings previously suggested based on archaeological and genetic data coincide with the presence of sufficiently wet corridors into Eurasia, indicating that palaeoclimatic conditions were an important factor in expansions out of Africa. Challenging environmental conditions in southwest Asia between windows of potential contact, interruptions of demographic influx from Africa, and possible competition with other hominins likely explain the demise of early colonists prior to the large-scale colonisation of the world beginning from ~65k years ago.

## Results and discussion

**Precipitation tolerance required for out-of-Africa exits.** Until recently, quasi-time-continuous palaeoclimate reconstructions were only available for the last 125k years, either from global circulation models such as HadCM3[28], or from simpler earth

models of intermediate complexity[26]. Here we used a recently developed emulator for the HadCM3 model to generate high-resolution climatologies of the last 300k years at 1k-year time intervals, downscaled to ~0.5°, and bias-corrected to match observed climatic conditions (Methods; Supplementary Movie 1). We then combined these reconstructions with simulations of annual palaeoclimatic variability to refine the temporal resolution of the climate maps down to a decadal scale (Methods). We consider two separate climate variables relevant for the survival of *Homo sapiens* in the region: annual precipitation, and aridity. For the latter, we used the Köppen aridity index[29], which incorporates both precipitation and temperature, and has been suggested as the most reliable aridity metric in palaeo-contexts[30]. We focus on precipitation and aridity as these were the ecologically limiting factors in the region and therefore likely the key climatic constraints of the ability of early modern humans to sustain themselves, by hunting and gathering animal- and plant-based food and collecting drinking water, during migrations out of Africa[31,32]. We consider two possible routes into Eurasia, through the Nile-Sinai-Land Bridge and the Strait of Bab-el-Mandeb, commonly denoted the northern and southern route, respectively[33]. We started by estimating, for each decade, the minimum requirement of annual precipitation that *Homo sapiens* would have had to withstand in order to be able to travel out of Africa. To estimate this value, we considered within each of the two general routes all specific geographic paths along which humans could have reached Eurasia with a given threshold of low precipitation, and then determined the lowest precipitation level for which a connected path out of Africa existed (Methods). We separately conducted an analogous analysis based on aridity. For our estimation, we assumed the Nile delta to be crossable at all times irrespective of the local precipitation and aridity levels. Modelling other features such as lakes and smaller rivers[31,34] over time is challenging; however, such features would have most likely occurred in areas wetter than the relatively low precipitation and aridity thresholds of human habitation. Furthermore, in our analysis of the southern route, we assumed the strait of Bab e-Mandeb to always be crossable. Whether indeed the necessary seafaring technology was available and used remains an open question, and we return to this issue later on. If maritime travel was in principle possible, then the difficulty of the voyage across the Bab e-Mandeb strait would have depended on sea level, which can change the width of the strait from 4 to over 20 km (Methods; Fig. 1b). Figure 1 shows the estimated precipitation tolerances that would have been required by *Homo sapiens* in the past 300k years in order for a migration out of Africa to be climatically feasible. Analogous results for Köppen aridity are qualitatively very similar (Supplementary Fig. 1a, b).

**Hunter-gatherer precipitation tolerance.** Periods during which climatic conditions would have allowed *Homo sapiens* to leave Africa for Eurasia can now be estimated by combining the inferred tolerance requirements over time with actual tolerance thresholds of early humans. To establish a plausible threshold to low precipitation, we first inspected the distribution of contemporary hunter-gatherers from a large anthropological dataset[35], which has previously been used to investigate spatial dynamics of early humans in response to environmental conditions[27,35–37]. Excluding three populations known to reside close to freshwater sources, there is a precipitation threshold around 90 mm of rainfall per year, below which no huntergatherers are recorded (Fig. 2a). This level also coincides with the minimum amount of precipitation that can sustain a grazer community[38] (Fig. 2a), and is near the switch from desert to xeric shrubland[39]. Present-day areas in northern Africa and the Arabian Peninsula experiencing

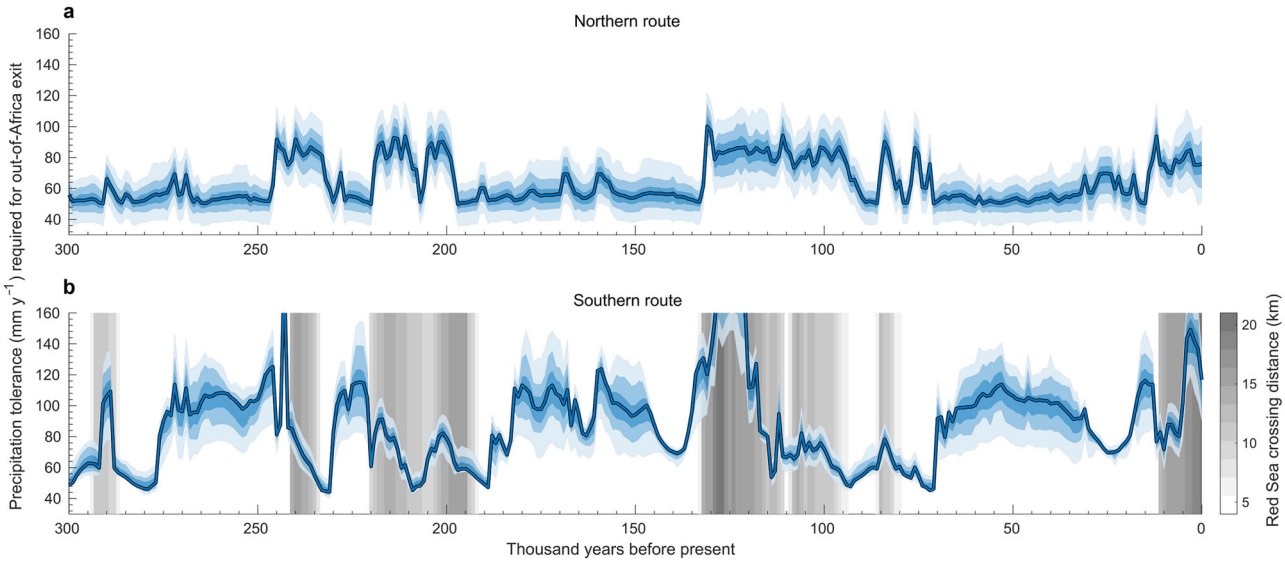

**Fig. 1 Tolerance to low precipitation that would have been required for *Homo sapiens* to leave the African continent in the past 300k years.** Blue lines represent minimum precipitation levels for which a connected path between Africa and Eurasia along the **a** northern **b** southern route existed, i.e. the minimum levels that humans would have had to withstand for a successful exit along these routes, based on climatological normals at 1k-year time steps (Methods). Thus, higher values correspond to a more favourable climate along the routes. Blue shades represent the 10–90th percentiles of the minimum precipitation tolerance required for a successful exit based on decadal scale climate (Methods). Grey shades in **b** represent the minimum distance needed to continuously cover on water to reach the Arabian Peninsula from Africa.

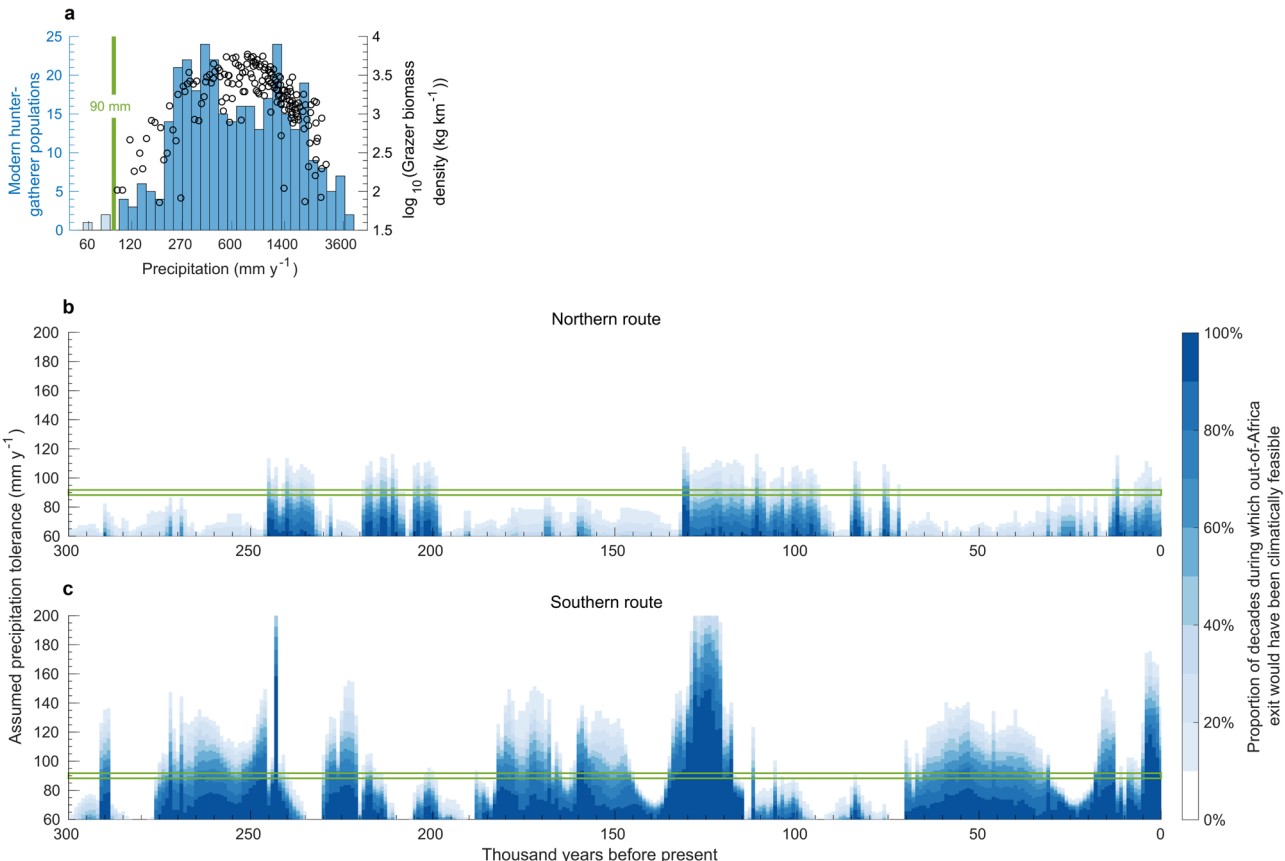

**Fig. 2 Climatic Windows of opportunity out of Africa. a** Distribution of modern hunter-gatherer populations (histograms)[36] under different precipitation levels and in relation to grazer biomass density[39] (black markers). Transparent bins correspond to populations located in close vicinity of a water source, which are not considered to be constrained by precipitation. **b**, **c** Percentage of decades within a given millennium (*x*-axis) during which a connected path between Africa and Eurasia along the **b** northern and **c** southern route existed for a given hunter-gatherer tolerance to low precipitation (*y*-axis). The green band represents our empirically estimated threshold of 90 mm y$^{-1}$.

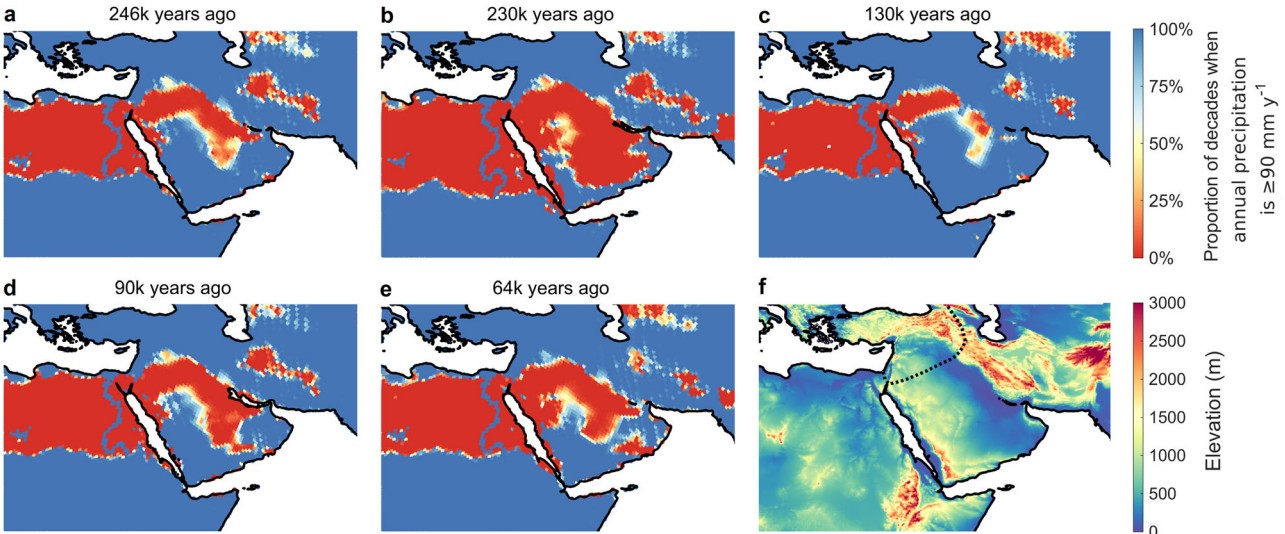

**Fig. 3 Environmental conditions in Northwest Africa and on the Arabian Peninsula.** Maps (**a**–**e**) illustrate reconstructed precipitation conditions at different points in time. Dark blue areas are estimated to have been sufficiently wet to support human persistence during most of the millennium shown, red areas were likely long-term unsuitable for human persistence, and orange, yellow, and light blue areas were inhabitable during intermittent periods. **a**, **c**, **d**, **e** correspond to key possible exit timings based on archaeological or genetic evidence while **b** exemplifies challenging conditions between windows of opportunity. The dotted line in the elevation map (**f**) represents the reconstructed Neanderthal range at 120k years ago[2]. Inhabitability maps for each millennium, and the analogous, very similar, maps based on Köppen aridity are shown in Supplementary Movie 2.

this level of rainfall are too dry for continuous grass cover, but allow for scattered reeds, grasses, and small shrubs, or pseudo-savannas characterised by patches of shrubs and herbs between scattered trees[40]. This vegetation can support several mammals adapted to very arid environments that would have been potential prey of *Homo sapiens*, including species of antelope, gazelle, fox, cat, shrew, and rodent[41].

**Climatic windows of opportunity out of Africa.** Based on our threshold estimate of 90 mm y$^{-1}$, there were a number of windows during the past 300k years when either northern or southern expansions out of Africa would have been climatically feasible (Fig. 2b,c, green band). Prior to the last interglacial period, the Nile-Sinai-Land Bridge would have been crossable at several time intervals between 246k and 200k years ago (cf. Fig. 3a). Following a reopening at 130k years ago (cf. Fig. 3c), exits would have been intermittently possible until 96k years ago (cf. Fig. 3d), and again later around 78k and 67k years ago. After that, this route likely remained closed until the wet Holocene. Provided that maritime travel was in principle possible, climatic conditions would have made southern exits feasible for a substantial proportion of the last 300k years. Before the last interglacial, there were three extended intervals of sufficient rainfall paired with relatively low sea level, from 275k to 242k years ago (cf. Fig. 3a), from 230k to 221k years ago, and from 182k to 145k years ago. During most of the following window from 135k to 115k years ago (cf. Fig. 3c), sea levels were particularly high, except at its beginning 135k years ago. This date is close to the proposed timing of an early northern exit; thus, if migration did occur, southern migrants might have encountered their northern counterparts on the Arabian Peninsula. Following a long period when the southern route was blocked, there was a sizeable window of sufficiently wet climate between 65k and 30k years ago (cf. Fig. 3e). Further connections existed just after the Last Glacial Maximum, and during the mid-Holocene, consistent with evidence of Eurasian backflow into Africa[42]. A threshold analogous to our estimate for precipitation exists for a Köppen aridity level around 1.7 based on the contemporary hunter-gatherer data

(Supplementary Fig. 2a), and the inferred periods of climatic connectivity between Africa and Eurasia are almost identical to those estimated for precipitation (Supplementary Fig. 2b,c).

Our reconstructions suggest that there were several windows of suitable climate along either of the two possible dispersal routes that could have allowed the expansion of *Homo sapiens* out of Africa. Some of these windows predate the earliest remains outside of Africa, but are entirely compatible with genetic dating of introgression from *Homo sapiens* into Neanderthal, sometime between 250k and 130k years ago[16,17], and recent dating of material from Israel to possibly 194k years ago[7] and from Greece to 210k years ago[10]. Migrations into Eurasia were also likely feasible along both routes during the last interglacial period, when archaeological evidence points to a more sizeable exit. Two distinct scenarios for an exit around 65k years ago, the time that has been long suggested as the main moment of expansion out of the African continent based on archaeological and genetic evidence, are compatible with our estimates. This timing marks both the point shortly after which the northern route last was open before a period of 40k years of unsuitable climate, and the point at which the southern route first reopened for an extended period since the last interglacial period. The latter scenario has been a subject of debate based on the empirical palaeoenvironmental record[43], with conclusions ranging from the Arabian Peninsula being continually too arid for human migration[23], to intermittent wet intervals[44,45], and extended pluvial periods[46] during marine isotope stage 3 (57–29k years ago). In any case, these inferences are not directly comparable with our results, both because several empirical proxies are not suitable for detecting rainfall of the small magnitude considered here (e.g. speleothems[47]), and because, for each route, the specific path out of Africa that requires the least tolerance to low precipitation out of all possible paths varies over time, as does the geographic location of its driest segment, whose rainfall level is shown in Fig. 1; thus, our estimates would not be expected to necessarily display the same patterns over time as a localised empirical climate reconstruction.

**Sensitivity of climatic windows to precipitation tolerance.** Assuming that precipitation and aridity tolerance thresholds

derived from contemporary hunter-gatherer data can be used as proxies for those of early modern humans is not without limitations. Ethnographically documented populations are not uniformly distributed across the world, residing predominantly in north America and Australia, south America, sub-Saharan Africa, and South and Southeast Asia[35], where both climatic and soil hydrological conditions can differ significantly from those in northern Africa and southwest Asia. In addition, technological differences, such as in the ability to store and transport water[49], may imply higher threshold levels (i.e. a requirement for higher levels of rainfall) for early hunter-gatherers. While currently available evidence may not allow us to quantitatively refine the tolerance thresholds derived from contemporary data, we can investigate the effect of different thresholds on the resulting windows of climatic connectivity between Africa and Eurasia (Fig. 2b,c). Along the northern route, our data suggest that precipitation tolerance thresholds above 110 mm y$^{-1}$ would have likely allowed for expansions out of Africa within the previously estimated windows only during shorter intervals with rainfall levels above the millennium-scale average. In this scenario, conditions would have been most favourable during the last interglacial period (around 130k years ago). For tolerance thresholds above 130 mm y$^{-1}$, migration would have likely been very challenging and restricted to unusually wet intervals. The southern route would have provided more scope for higher precipitation tolerance levels. Thresholds up to 200 mm y$^{-1}$ would have provided opportunities to leave Africa during the last interglacial period. Between this point and the wet Holocene, tolerance levels of up to 130 mm y$^{-1}$ would have likely allowed for migrations into Eurasia during favourable intervals between 65k and 55k years ago.

**Difficulty of crossing the Red Sea.** In addition to climatic constraints, the requirement to cross the strait of the Bab e-Mandeb would have posed a key challenge to migration along the southern route. Whether early hunter-gatherers ventured across the Red Sea, as has been suggested notably based on genetic evidence[48,50], remains a subject of debate given the very limited archaeological support for this scenario[51]. During periods of low sea level, the Arabian Peninsula would have been visible from present-day Djibouti and southeast Eritrea, and crossing the strait at these times may not have required sophisticated boats or seafaring skills[52]. However, whilst it is likely that humans lived on the western Red Sea coast and used marine food resources[53], direct evidence of boats and maritime travel remains to be found. In addition, although technological similarities have been suggested between some sites on the Arabian Peninsula and in Northeast Africa[54], other sites show no such relationship[55]. Caution should therefore be taken before interpreting the more favourable climate along the southern route suggested by our data as evidence for a southern exit; instead, expanding and reconciling genetic and archaeological lines of evidence remains crucial for clarifying the role of the Bab e-Mandeb strait in expansions out of Africa.

**Short-term climatic variability and early-human demography.** Our analysis of the climatic feasibility of *Homo sapiens* leaving the African continent in the last 300k years is based on decadal-scale variability of annual precipitation and aridity. Whilst neither empirical nor simulation-based approaches currently appear capable of reconstructing climatic conditions at a higher temporal resolution across the same time period and geographical area without compromising robustness, it is important to bear in mind that short-term climatic variability can play an important role for human population dynamics. Storms and monsoon rains followed by extended dry spells would have posed different challenges than the same total rainfall spread out over a long period.

Our results provide estimates of when migration between Africa and Eurasia would have been climatically feasible for *Homo sapiens*, not whether these potential windows of opportunity were seized. This would require combining our data with human dispersal models that explicitly simulate spatio-temporal population dynamics, which we have not attempted here. Uncertainties associated with such an approach would likely be substantial given that current estimates of parameters relevant to demographic processes, such as population growth rates and dispersal speeds, range across several orders of magnitude[26,27]. Likewise, important questions such as to what extent movement patterns of early modern humans were directional or random, and how they varied in different environments and in response to changes in population size, largely lack quantitative answers. Consolidating anthropological, archaeological, and genetic data would seem as the most promising avenue towards reducing existing uncertainties.

**Failure of early waves to settle in Eurasia.** While archaeological and genetic data strongly suggest that *Homo sapiens* expanded its range towards Eurasia at least once prior to the large-scale colonisation wave beginning around 65k years ago, the reason for its initial failure to permanently settle outside Africa is less clear. Migration beyond the Arabian Peninsula would have been predicated on the ability to cross the Taurus-Zagros Mountain range while competing with Neanderthals in the north (Fig. 3f), which has previously been argued to have limited human expansions during the last interglacial period[4,56], and possibly other hominins, such as Denisovans (whose geographic range is unknown but likely covered a large portion of East Asia[57]), in the east. In addition, our reconstructions suggest that climatically favourable intervals along both routes were frequently interrupted by periods of rainfall insufficient to support humans (Fig. 2b,c; Fig. 3b), which would have effectively isolated any of the earlier colonists that might have made it out of Africa. With a lack of demographic influx from further migration out of Africa, remnant populations on the Arabian Peninsula would have been susceptible to stochastic local extinctions driven by climatic fluctuations[58]. This constraint would have been less important along the southern route during the unprecedentedly long period of largely favourable climate between 65k and 30k years ago (Fig. 2c), provided that maritime travel across a then 4 km wide strait of the Bab e-Mandeb made migration along this route possible. This long window would have provided ideal preconditions for a successful large-scale dispersal, allowing for a regular demographic influx from Africa that would have stabilised populations on the Arabian Peninsula, thus facilitating further expansion of *Homo sapiens* into Eurasia. These dynamics would have complemented technological, economic, social, and cognitive changes in human societies[4,56,59], which, possibly combined with the decline of Neanderthal[60,61], very probably accounted for the success of the late exit in the subsequent colonisation of Eurasia by *Homo sapiens*.

## Methods

### Late Quaternary climate reconstructions
*Precipitation.* Our reconstructions of Late Quaternary precipitation are based on outputs from a statistical emulator of the HadCM3 general circulation model[62]. The emulator was developed using 72 3.75° × 2.5° resolution snapshot climate simulations of HadCM3, covering the last 120k years and in 2k year time steps from 120k to 22k years ago and 1k-year time steps from 21k years ago to the present, where each time slice represents climatic conditions averaged across a 30-year post-spin-up period[28,63]. The emulator is based on grid-cell-specific linear regressions between the local time series of HadCM3 climate data and four time-dependent forcings, given by the mean global atmospheric $CO_2$ concentration and three orbital parameters: eccentricity, obliquity, and precession. The values of these four predictors are known well beyond the last 120k years; thus, applying them to the calibrated grid-cell-specific linear regressions allows for the statistical

extrapolation of global climate up to 800k years into the past[62]. The emulated climate data have been shown to correspond closely to the original HadCM3 simulations for the last 120k years, and to match long-term empirical climate reconstructions well[62].

Here, we used precipitation data from the emulator, denoted $\bar{\mathbf{P}}_{\text{HadCM3}_{\text{em}}}(t)$, of the last 300k years at 1k-year time steps, $t \in \mathbf{T}_{300k}$. The data were spatially downscaled from their native $3.75° \times 2.5°$ grid resolution, and subsequently bias-corrected, in two steps, similar to the approach described in ref. [64], whose description we follow here. Both steps use variations of the delta method[65], under which a high-resolution, bias-corrected reconstruction of precipitation at sometime $t$ is obtained by applying the difference between lower-resolution present-day simulated and high-resolution present-day observed climate—the correction term—to the simulated climate at time $t$. The delta method has been used to downscale and bias-correct palaeoclimate simulations before (e.g. for the WorldClim database[66]), and, despite its conceptual simplicity, has been shown to outperform alternative methods commonly used for downscaling and bias-correction[67].

A key limitation of the delta method is that it assumes the present-day correction term to be representative of past correction terms. This assumption is substantially relaxed in the dynamic delta method used in the first step of our approach to downscale $\bar{\mathbf{P}}_{\text{HadCM3}_{\text{em}}}(t)$ to a ~1° resolution. This method involves the use of a set of high-resolution climate simulations that were run for a smaller but climatically diverse subset of $\mathbf{T}_{300k}$. Simulations at this resolution are computationally very expensive, and therefore running substantially larger sets of simulations is not feasible; however, these selected data can be very effectively used to generate a suitable time-dependent correction term for each $t \in \mathbf{T}_{300k}$. In this way, we can increase the resolution of the original climate simulations by a factor of ~9, while simultaneously allowing for the temporal variability of the correction term. In the following, we describe the approach in detail.

We used high-resolution precipitation simulations from the HadAM3H model[63], generated for the last 21,000 years in 9 snapshots (2k year time intervals from 12k to 6k years ago, and 3k year time intervals otherwise) at a $1.25° \times .83°$ grid resolution, denoted $\bar{\mathbf{P}}_{\text{HadAM3H}}(t)$, where $t \in \mathbf{T}_{21k}$ represents the nine time slices for which simulations are available. These data were used to downscale $\bar{\mathbf{P}}_{\text{HadCM3}_{\text{em}}}(t)$ to a $1.25° \times 0.83°$ resolution by means of the multiplicative dynamic delta method, yielding

$$\bar{\mathbf{P}}_{\sim 1°}(t) \stackrel{\text{def}}{=} \bar{\mathbf{P}}^{\boxplus}_{\text{HadCM3}_{\text{em}}}(t) \cdot \frac{\bar{\mathbf{P}}_{\text{HadAM3H}}(\hat{t})}{\bar{\mathbf{P}}^{\boxplus}_{\text{HadCM3}_{\text{em}}}(\hat{t})}. \quad (1)$$

The $\boxplus$-notation indicates that the coarser-resolution data were interpolated to the grid of the higher-resolution data, for which we used an Akima cubic Hermite interpolant[68], which, unlike the bilinear interpolation, is continuously differentiable but, unlike the bicubic interpolation, avoids overshoots. The time $\hat{t} \in \mathbf{T}_{21k}$ is chosen as the time at which climate was, in a sense specified below, close to that at time $t \in \mathbf{T}_{300k}$. In contrast to the classical delta method (for which $\hat{t} = 0$ for all $t$), this approach does not assume that the resolution correction term, $\left(\frac{\bar{\mathbf{P}}_{\text{HadAM3H}}(\hat{t})}{\bar{\mathbf{P}}^{\boxplus}_{\text{HadCM3}_{\text{em}}}(\hat{t})}\right)$, is constant over time. Instead, the finescale heterogeneities that are applied to the coarser-resolution $\bar{\mathbf{P}}_{\text{HadCM3}_{\text{em}}}(t)$ are chosen from the wide range of patterns simulated for the last 21k years. The strength of the approach lies in the fact that the last 21k years account for a substantial portion of the glacial-interglacial range of climatic conditions present during the whole Late Quaternary. Following ref. [64], we used global $CO_2$, a key indicator of the global climatic state, as the metric according to which $\hat{t}$ is chosen; i.e. among the times for which HadAM3H simulations are available, $\hat{t}$ is the time at which global $CO_2$ was closest to the respective value at the time of interest, $t$.

In the second step of our approach, we used the classical multiplicative delta method to bias-correct and further downscale $\mathbf{P}_{\sim 1°}(t)$ to a hexagonal grid[69] with an internode spacing of ~55 km (~0.5°),

$$\bar{\mathbf{P}}_{\sim 0.5°}(t) \stackrel{\text{def}}{=} \bar{\mathbf{P}}^{\boxplus}_{\sim 1°}(t) \cdot \frac{\bar{\mathbf{P}}_{\text{obs}}(0)}{\bar{\mathbf{P}}^{\boxplus}_{\sim 1°}(0)}, \quad (2)$$

where $\mathbf{P}_{\text{obs}}(0)$ denotes present-era (1960–1990) observed precipitation[70].

We reconstructed land configurations for the last 300k years using present-day elevation[71] and a time series of Red Sea sea level[72]. For locations that are currently below sea level, the delta method does not work. For these locations, precipitation was extrapolated using a inverse distance weighting approach. With the exception of a brief window from 124–126k years ago, sea level in the past was lower than it is today; thus, present-day coastal patterns are spatially extended as coastlines move, but not removed. For all $t \in \mathbf{T}_{300k}$, maps of annual precipitation $\bar{\mathbf{P}}_{\sim 0.5°}(t)$ with the appropriate land configuration are available as Supplementary Movie 1.

Based on these data representing 30-year climatological normals at 1k-year time steps between 300k years ago and the present, we generated, for each millennium, 100 maps representing 10-year average climatologies as follows. We used $3.75° \times 2.5°$ climate simulations from the HadCM3B-M2.1 model, providing a 1000-years-long annual time series of annual precipitation for each millennium between 21k years ago and the present[73]. Millennia were simulated in parallel; thus, the 1000-years-long time series representing each millennium is in itself continuous, but the beginnings and ends of the time series of successive millennia generally do not coincide. For $t \in \mathbf{T}_{21k}$, we denote the available 1000 successive maps of annual

precipitation by $\mathbf{P}^{(1)}_{\text{HM}}(t), \dots, \mathbf{P}^{(1000)}_{\text{HM}}(t)$. We used these data to compute the relative deviation of the climatic average of each decade within a given millennium, and the climatic average of the 30-year period containing the specific decade as

$$\boldsymbol{\epsilon}^{(d)}_{\text{HM}}(t) \stackrel{\text{def}}{=} \frac{\sum_{i=1+(d-1)\cdot 10}^{d\cdot 10} \mathbf{P}^{(i)}_{\text{HM}}(t)}{\sum_{n=1+(d-2)\cdot 10}^{(d+1)\cdot 10} \mathbf{P}^{(n)}_{\text{HM}}(t)}, \quad d = 1, \dots, 100 \quad (3)$$

Finally, we applied these ratios of 10-year to 30-year climatic averages to the previously derived 1k-year time step climatologies to obtain, for each $t \in \mathbf{T}_{300k}$, 100 sets of 10-year average annual precipitation,

$$\mathbf{P}^{(d)}_{\sim 0.5°}(t) \stackrel{\text{def}}{=} \bar{\mathbf{P}}_{\sim 0.5°}(t) \cdot \boldsymbol{\epsilon}^{(d),\boxplus}_{\text{HM}}(\hat{t}), \quad d = 1, \dots, 100 \quad (4)$$

where, analogous to our approach in Eq. (1), $\boxplus$ denotes the interpolation to the ~55 km hexagonal grid, and where $\hat{t}$ is chosen as the time at which global $CO_2$ was closest to the respective value at time $t$.

*Aridity.* The Köppen aridity index used here is defined as the ratio of annual precipitation (in mm) to the sum of mean annual temperature (in °C) and a constant of 33 °C (cf. Eq. (8)). This measure of aridity was found to be the most reliable one of a set of alternative indices in palaeoclimate contexts[30].

Decadal-scale mean annual temperature data between 300k years ago and the present were created using analogous methods to those previously applied to reconstruct precipitation. $3.75° \times 2.5°$ resolution emulator-derived simulations of mean annual temperature of the past 300k years at 1k time steps[62], denoted $\bar{\mathbf{T}}_{\text{HadCM3}_{\text{em}}}(t)$, were first downscaled by means of the additive dynamic delta method, using $1.25° \times 0.83°$ HadAM3H simulations of mean annual temperature of the past 21k years, denoted $\bar{\mathbf{T}}_{\text{HadAM3H}}(t)$, yielding, analogous to Eq. (1),

$$\bar{\mathbf{T}}_{\sim 1°}(t) \stackrel{\text{def}}{=} \bar{\mathbf{T}}^{\boxplus}_{\text{HadCM3}_{\text{em}}}(t) + \left(\bar{\mathbf{T}}_{\text{HadAM3H}}(\hat{t}) - \bar{\mathbf{T}}^{\boxplus}_{\text{HadCM3}_{\text{em}}}(\hat{t})\right). \quad (5)$$

Analogous to Eq. (2), Next, present-day observed mean annual temperature, $\bar{\mathbf{T}}_{\text{obs}}(0)$, was used to further downscale and bias-correct the data by means of the additive delta method to obtain

$$\bar{\mathbf{T}}_{\sim 0.5°}(t) \stackrel{\text{def}}{=} \bar{\mathbf{T}}^{\boxplus}_{\sim 1°}(t) + \left(\bar{\mathbf{T}}_{\text{obs}}(0) - \bar{\mathbf{T}}^{\boxplus}_{\sim 1°}(\hat{t})\right). \quad (6)$$

For all $t \in \mathbf{T}_{300k}$, maps of mean annual temperature $\bar{\mathbf{T}}_{\sim 0.5°}(t)$ with the appropriate land configuration are available as Supplementary Movie 1.

Finally, we incorporated HadCM3B-M2 simulations of mean annual temperature of the past 21k years, $\mathbf{T}^{(1)}_{\text{HM}}(t), \dots, \mathbf{T}^{(1000)}_{\text{HM}}(t)$ for $t \in \mathbf{T}_{21k}$, to obtain 10-year average mean annual temperature,

$$\mathbf{T}^{(d)}_{\sim 0.5°}(t) \stackrel{\text{def}}{=} \bar{\mathbf{T}}_{\sim 0.5°}(t) + \left(\sum_{i=1+(d-1)\cdot 10}^{d\cdot 10} \bar{\mathbf{T}}^{(i)}_{\text{HM}}(t) - \sum_{n=1+(d-2)\cdot 10}^{(d+1)\cdot 10} \bar{\mathbf{T}}^{(n)}_{\text{HM}}(t)\right)^{\boxplus}, \\ d = 1, \dots, 100 \quad (7)$$

Based on these data, the Köppen aridity index at the same spatial and temporal resolution is calculated as

$$\mathbf{A}^{(d)}_{\sim 0.5°}(t) \stackrel{\text{def}}{=} \frac{\mathbf{P}^{(d)}_{\sim 0.5°}(t)}{\mathbf{T}^{(d)}_{\sim 0.5°}(t) + 33}. \quad (8)$$

*Comparison with empirical proxies*

**Long-term proxy records**
Long-term proxy records allow us to assess whether simulations capture key qualitative dynamics observed in the empirical data. The lack of direct long-term time series reconstructions of annual precipitation and mean annual temperature makes it necessary to use proxies related to these two climate variables. Proxies providing temporal coverage beyond the last glacial maximum are not only extremely sparse in North Africa and Southwest Asia, but even the few records that exist are affected by environmental factors other than the specific climate variables considered here. For example, reconstructions of past wetness and aridity use proxies that reflect not only rainfall conditions but also the interaction of precipitation with other local and non-local hydro-climatic variables, e.g. river discharge or hydrological catchment across a larger area. Here, we have not attempted to correct for such processes, but assumed that the simulated climate at the site where the empirical record was taken provide a suitable approximation of the potentially broader climatic conditions relevant for the proxy data. Realistic climate simulations would therefore be expected to match major qualitative trends of the empirical records, rather than exhibit a perfect correlation with the data. We compared our precipitation simulations against three long-term humidity-related empirical proxies (Fig. 4a). Proxy 1[74] provides a time series of Dead Sea lake levels, for which wet and dry periods are associated with high-stand and low-stand conditions, respectively. Proxy 2[19] from the southern tip of the Arabian Peninsula was obtained from a marine sediment core that allows for reconstructing past changes in aridity over land from the stable hydrogen isotopic composition of leaf waxes ($\delta D_{\text{wax}}$). Proxy 3[18] is an XRF-derived humidity index from a core near the Northwest African coast. Temperature simulations were compared against two long-term records of $\delta^{18}O$, which varies over time as a result of temperature fluctuations (in addition to other factors), from the Peqiin and Soreq caves in Northern Israel[75] (Fig. 4e). Overall, the simulated data capture key phases observed in the empirical records well for both precipitation (Fig. 4b–d) and temperature proxies (Fig. 4f–h).

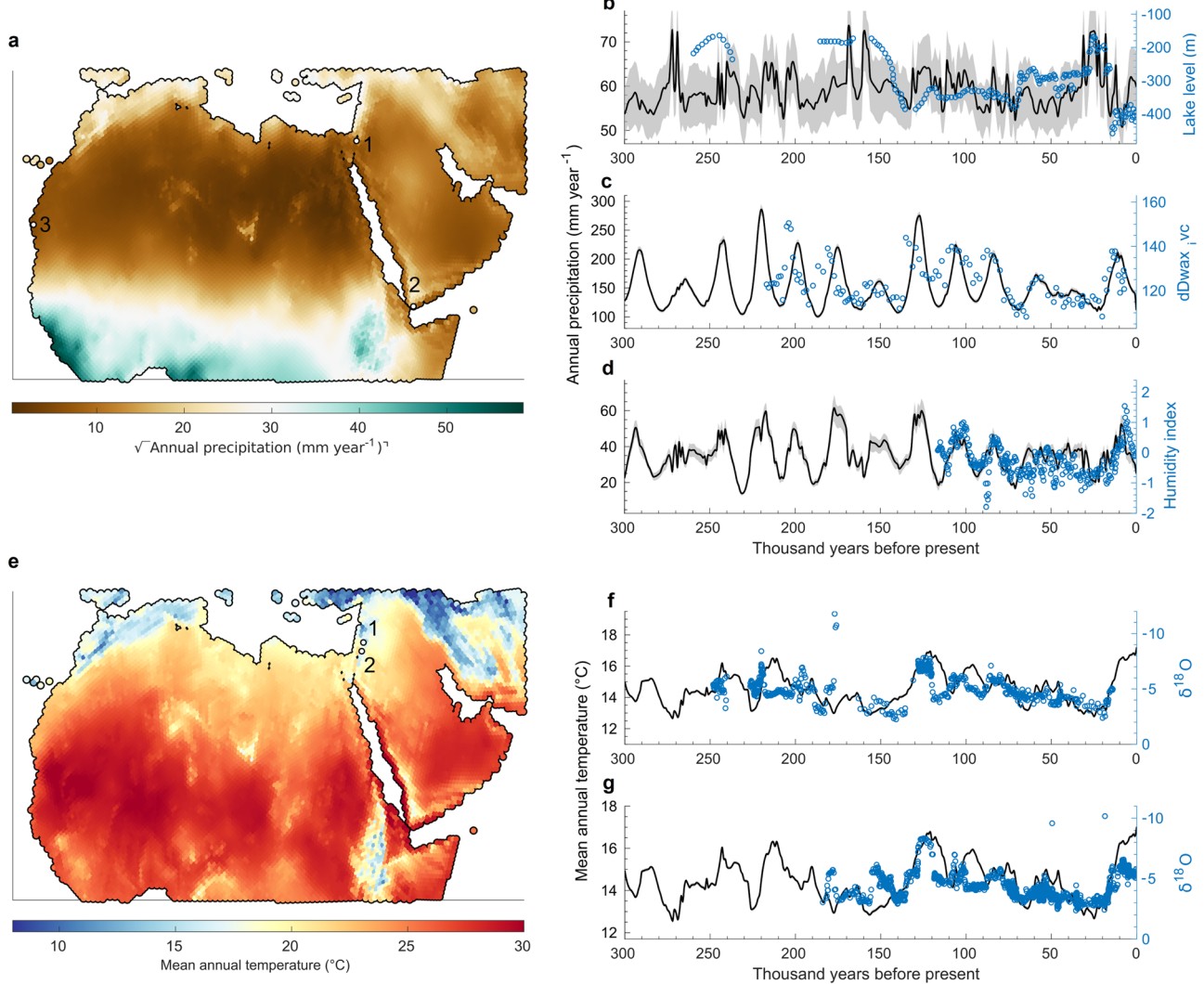

**Fig. 4 Comparison of our data to long-term proxy records. a** Geographical locations of empirical proxies on a map of present-day annual precipitation. **b–d** Comparisons of simulated annual precipitation against the three wetness proxies. **e** Geographical locations of empirical proxies on a map of present-day mean annual temperature. **f, g** Comparisons of simulated mean annual temperature against the two δ¹⁸O records. Black lines represent the simulated climatological normals at 1k-year intervals (Eqs. (2) and (6)), grey shades represent the 10th and 90th percentile of the decadal simulations (n = 100; Eqs. (4) and (7)).

### Pollen-based reconstructions

Pollen records used to empirically reconstruct past climate do not reach as far back in time as the above-described proxy records and are not available at the same temporal resolution; however, in contrast to those proxies, they can be used to quantitatively estimate local annual precipitation and mean annual temperature directly. Here, we used the dataset of pollen-based reconstructions of precipitation and temperature for the mid-Holocene (6k years ago) and the last glacial maximum (21k years ago)[76] (Fig. 5a). Our precipitation and temperature data are overall in good agreement with the empirical reconstructions (Fig. 5b–e). During the mid-Holocene, our simulations suggest slightly less precipitation at low levels than most of the empirical records (Fig. 5d), while our data match the empirical reconstruction available from a very arid location during the last glacial maximum very well (Fig. 5e).

### Interglacial palaeolakes on the Arabian Peninsula

Finally, we plotted time series of our precipitation simulations in three locations in which palaeolakes have been dated to the last interglacial period, following the approach in ref. [24], in which the authors tested whether their climate simulations predicted higher rainfall during the last interglacial period than at present at palaeolake sites on the Arabian Peninsula. Figure 6 shows the locations of three palaeolakes in the northeast (western Nefud near Taymal; proxy 1), the centre (at Khujaymah; proxy 2), and the southwest (at Saiwan; proxy 3) of the peninsula[24] (described in detail in refs. [23,77]), and our precipitation data in these locations. In two out of the three locations, our data predict that more rainfall occurred at the estimated timings of the palaeolakes than at any

point in time since; in the third location, slightly more rainfall than during the dated time interval is simulated only for a period around 8k years ago.

### Determining the minimum precipitation and aridity tolerance required for out-of-Africa exits

We denote by $\mathbf{X} = \{(\lambda_1, \phi_1), (\lambda_2, \phi_2), \dots\}$ the set of longitude and latitude coordinates of the hexagonal grid with an internode spacing of ~55 km (~0.5°)[69] that are contained in the longitude window [15°E, 70°E] and the latitude window [5°N, 43°N] (shown in Fig. 3). We denote by $\mathbf{E}$ the set of the present-day elevation values of the coordinates in $\mathbf{X}$ (in meters)[78], i.e. $\mathbf{E}(x_i)$ is a positive number in a point $x_i = (\lambda_i, \phi_i)$ if $x_i$ is currently above sea level, and negative if $x_i$ is currently below sea level. We denote by $s(t)$ the sea level (in meters) at the time $t \in \mathbf{T}_{300k}$ (where $\mathbf{T}_{300k}$ represents the last 300k years in 1k time steps), for which we used a long-term reconstruction of Red Sea sea level[72]. In particular, we have $s(0) = 0$ at present day. For each millennium $t \in \mathbf{T}_{300k}$, we denote by $\bar{\mathbf{X}}(t)$ the subset of points in $X$ that are above sea level:

$$\bar{\mathbf{X}}(t) \stackrel{\text{def}}{=} \{x \in \mathbf{X} : \mathbf{E}(x) > s(t)\} \qquad (9)$$

Based on the precipitation map $\mathbf{P}_{\sim 0.5°}^{(d)}(t)$ for a decade $d = 1, \dots, 100$ in millennium $t$ (Eq. (4)), and a given precipitation threshold value $p$ (in mm year⁻¹), we denote by $\bar{\bar{\mathbf{X}}}_p^{(d)}(t)$ the subset of $\bar{\mathbf{X}}(t)$ that would be suitable grid cells for humans

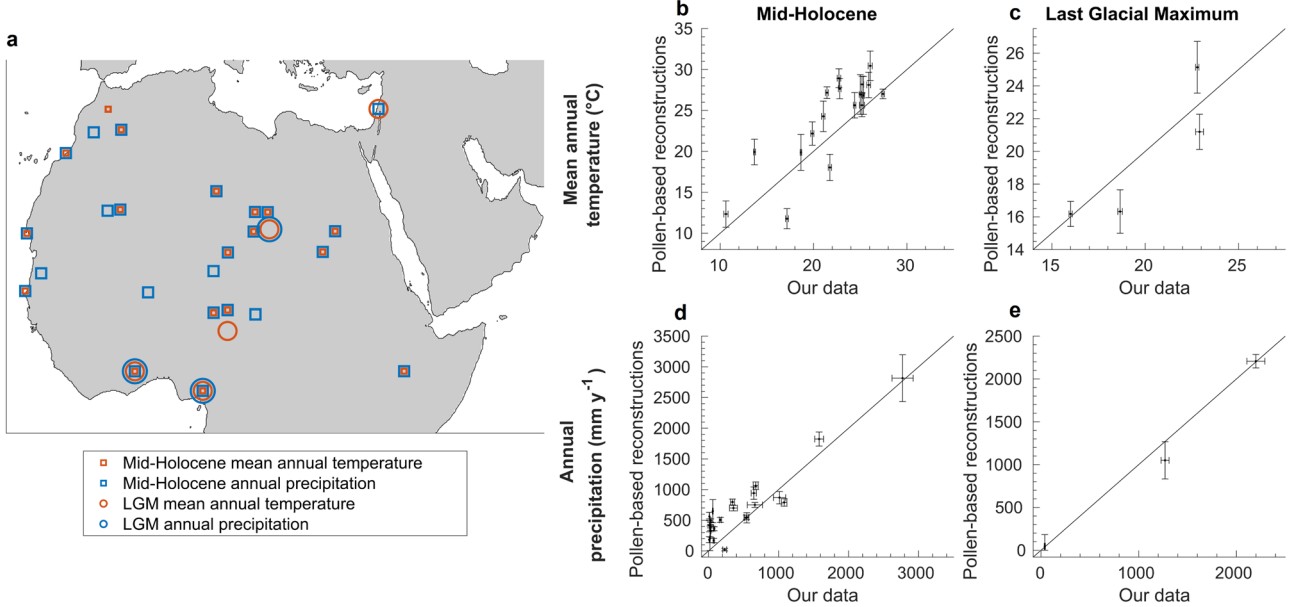

**Fig. 5 Comparison of our data to pollen-based climate reconstructions from the mid-Holocene and the last interglacial period. a** Geographical locations and timings of pollen records. **b–e** Comparisons of our data against empirical reconstructions. Vertical centre measures and error bars represent the empirical reconstructed values and their uncertainties, respectively; horizontal centre measures and error bars represent simulated climatological normals at 1k-year intervals (Eqs. (2) and (6)) and the 10th and 90th percentile of the simulated decadal data ($n = 100$; Eqs. (4) and (7)), respectively.

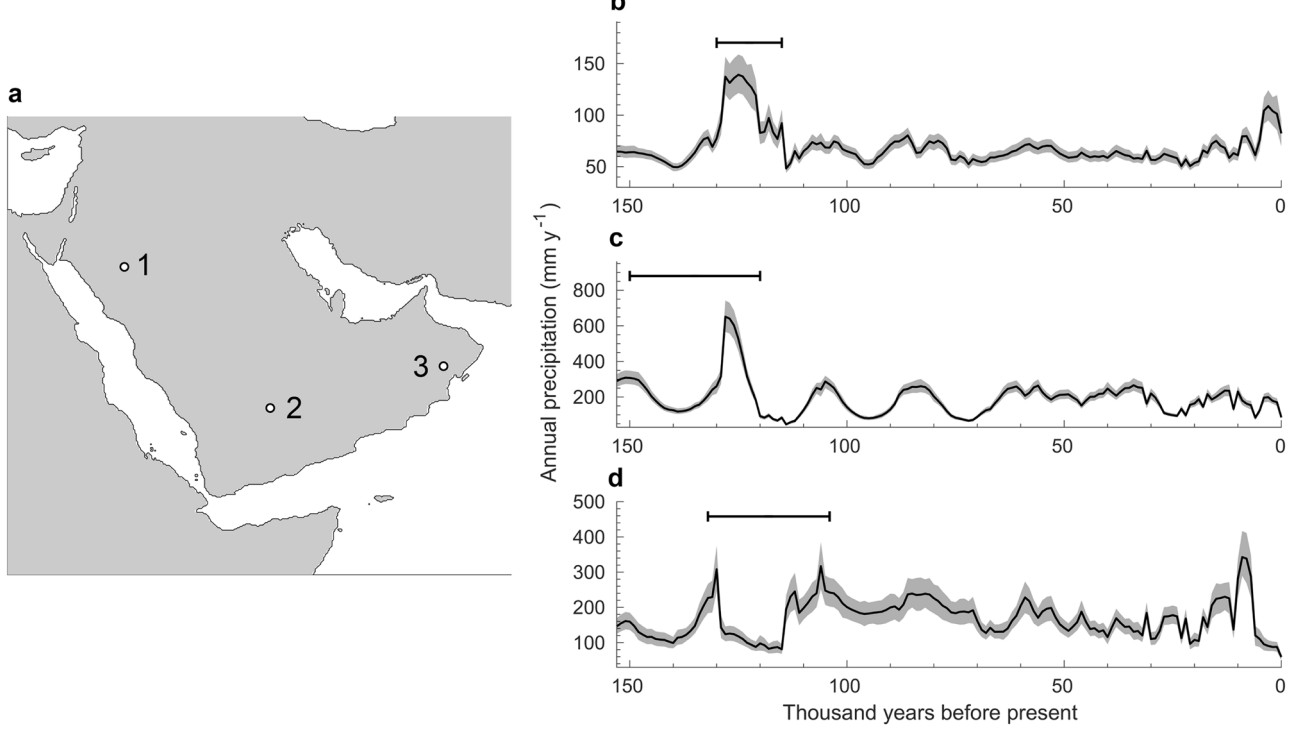

**Fig. 6 Comparison of our data against the dates of three palaeolakes on the Arabian peninsula. a** Geographical locations of the lakes. **b–d** Time series of our precipitation data. Black lines represent the simulated climatological normals at 1k-year intervals (Eqs. (2) and (6)), grey shades represent the 10th and 90th percentile of the decadal simulations ($n = 100$; Eq. (4)). Horizontal error bars represent the estimated dates of the lakes[24].

assuming that they cannot survive in areas where precipitation levels are below $p$:

$$\overline{\overline{\mathbf{X}}}_p^{(d)}(t) \stackrel{\text{def}}{=} \left\{ x \in \overline{\mathbf{X}}(t) : \mathbf{P}_{\sim 0.5°}^{(d)}(t) \geq p \right\} \quad (10)$$

We then determined whether there was a connected path in $\overline{\overline{\mathbf{X}}}_p^{(d)}(t)$ between an initial point, for which we used $x_{\text{start}} = (32.6°\text{E}, 10.2°\text{N})$, and any point in a set of coordinates outside of Africa, defined as $\mathbf{X}_{\text{end}} \stackrel{\text{def}}{=} \{(\lambda, \phi) \in \mathbf{X} : \lambda > 65°\text{E or } \phi > 37°\text{N}\}$.

This was defined to be the case if there was a finite sequence

$$x_{\text{start}} \rightarrow x_1 \rightarrow x_2 \rightarrow \ldots \rightarrow x_n \in \mathbf{X}_{\text{end}} \quad (11)$$

of points $x_i \in \overline{\overline{\mathbf{X}}}_p^{(d)}(t)$ such that the distance between any two successive points $x_i$ and $x_{i+1}$ was less or equal to the maximum internode spacing of the grid $X$. Based on this approach, the critical precipitation threshold below which no connected path exists for the precipitation map $\mathbf{P}_{\sim 0.5°}^{(d)}(t)$ was determined using the following

bisection method. Beginning with $\hat{p}_0 = 1000$ mm y$^{-1}$ and $\check{p}_0 = 0$ mm y$^{-1}$, for which a connected path between $x_{\text{start}}$ and $\mathbf{X}_{\text{end}}$ exists, respectively, for all and for no $t$ and $d$, the values $\hat{p}_k$ and $\check{p}_k$ were iteratively defined as

$$\left.\begin{array}{l} \check{p}_{k+1} \stackrel{\text{def}}{=} \frac{\hat{p}_k + \check{p}_k}{2} \\ \hat{p}_{k+1} \stackrel{\text{def}}{=} \hat{p}_k \end{array}\right\} \text{ if a connected path exists for } p = \frac{\hat{p}_k + \check{p}_k}{2}$$

$$\left.\begin{array}{l} \check{p}_{k+1} \stackrel{\text{def}}{=} \check{p}_k \\ \hat{p}_{k+1} \stackrel{\text{def}}{=} \frac{\hat{p}_k + \check{p}_k}{2} \end{array}\right\} \text{ else}$$ (12)

For all $k$, the sought critical precipitation threshold, denoted $p_{\text{crit}}^{(d)}(t)$, is bounded above by $\hat{p}_k$ and bounded below by $\check{p}_k$. For $k \to \infty$, both values converge to $p_{\text{crit}}^{(d)}(t)$. Here, we defined

$$p_{\text{crit}}^{(d)}(t) \stackrel{\text{def}}{=} \frac{\hat{p}_{10} + \check{p}_{10}}{2},$$ (13)

which lies within 1 mm y$^{-1}$ of the true limit value.

To specifically determine the precipitation tolerance required for a northern (Fig. 1a) or southern (Fig. 1b) exit, we rendered the passage of the respective other route impassable by removing appropriate cells from the grid. When investigating the southern route, we additionally assumed that no sea level and precipitation constraints applied within a ~40 km radius around the centre of the Bab al-Mandab strait.

For aridity, the procedure is identical, with the exception that $\overset{=(d)}{\mathbf{X}}_p(t)$ is defined based on the relevant aridity map, $\mathbf{A}_{\sim 0.5°}^{(d)}(t)$, and the value 4.0 is used for the initial upper threshold (denoted $\hat{p}_0$ above).

*Width of the Strait of Bab al-Mandab.* Similar to ref. [52], we reconstructed the minimum distance required to cover on water in order to reach the Arabian peninsula (present-day west coast of Yemen) from Africa (present-day Djibouti and southeast Eritrea). We used a 0.0083° (~1 km at the equator) map of elevation and bathymetry[78] and a time series of Red Sea sea level[72] to reconstruct very-high-resolution land masks for the last 300k years. For each point in time, we determined the set of connected land masses, and the distances between the closest points of any two land masses. The result can be graph-theoretically represented by a complete graph whose nodes represent connected land masses and whose edge weights correspond to the minimum distances between land masses. The path involving the minimum continuous distance on water was then determined by solving the minmax path problem whose solution is the path between the two nodes representing Africa and the Arabian Peninsula that minimises the maximum weight of any of its edges (Fig. 1b grey shades).

Analyses were conducted using Matlab R2019a[79].

**Reporting summary**. Further information on research design is available in the Nature Research Reporting Summary linked to this article.

## Data availability
All data associated with this study are available on the Open Science Framework (https://doi.org/10.17605/OSF.IO/NMS4F)[80].

## Code availability
All code associated with this study is available on the Open Science Framework (https://doi.org/10.17605/OSF.IO/NMS4F)[80].

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

## Acknowledgements

R.M.B., M.K., and A.M. were supported by ERC Consolidator Grant 647797 "LocalAdaptation". A.E. was supported by the European Union through Horizon 2020 research and innovation programme under grant number 810645.

## Author contributions

A.M., R.M.B., M.K., and A.E. conceived the study, interpreted the results and revised the manuscript. M.K and R.M.B. generated the palaeoclimate data. R.M.B. conducted the analysis. A.M. and R.M.B. wrote the manuscript.

## Competing interests

The authors declare no competing interests.
