## [Peer Review File · Nature Communications]

Reviewers' Comments:

Reviewer #1:

Remarks to the Author:

I have mixed feelings about this paper. In positive terms, it is exactly the kind of study that is needed, i.e. quantitative reconstruction of regional paleoenvironmental conditions in the context of debates about the timing and character of dispersals out of Africa. However, I think that there are significant problems with some of the specifics of the manuscript at it stands. The study has the potential be highly novel and interesting to researchers in multiple disciplines. I think if the analyses can be repeated with a more realistic water-tolerance level then this will be a highly significant and influential paper.

The authors response to reviewers on the previous version of the paper appear generally reasonable, yet they are also somewhat cosmetic and the substantial methods and conclusions of the paper remain the same. But by doing things like adding an aridity index, they have improved the paper...I just think that they have not gone far enough with the changes. Likewise, adding a 5% error buffer is good, but likewise I don't think it's enough.

Aims and methods:

If the aim is looking for a dispersal 'corridor', things could be much easier: the Nile Valley offers a corridor for much of the relevant time period. Plus given the importance of physical geography factors (e.g. a sea crossing being involved in the southern route) mean that talk of 'corridors' is rather illusory. I think the aim of the paper should be cast more as about reconstructing regional environmental conditions – an important goal, given that most current paleoenvironmental archives offer scattered and often poorly dated insights. Their model arguably tells us more about what we could call 'survivability' than 'dispersability'. How conditions in a particular area might relate to dispersal routes needs, in my opinion, rather more nuanced consideration.

The key specific point here is about whether early humans crossed the southern Red Sea. Crossing a sea with strong currents is a strange 'corridor' for a terrestrially adapted animal (which does not mean it was not used, but just it seems a rather strange 'corridor'). If we see the dispersal of human populations across thousands of kilometers of challenging landscapes as an 'event' (as in the 'out of Africa event' that they, following common terminology in genetics, use), then describing a 'corridor' makes sense, but in the applied context of ecology, environment and human social dynamics it is a rather problematic concept. While I think the concepts of corridors as applied here is dubious, the one sense in which it is perhaps useful – that is river systems which cross the arid areas (e.g. Drake et al., 2011 in PNAS) – is actually ignored here. I understand why that is so, but they should then be more explicit about what they are doing being about reconstructing regional scale patterns, not actual corridors.

Survivability and dispersibility:

One of the things the authors have changed in this new version of the paper is an attempt to more explicitly deal with the highly variable nature of climate in these areas. In my opinion this is absolutely

fundamental. It doesn't really matter if ten years have high enough rainfall for survival if there is not a drop in year eleven. The authors suggest in the response to reviewers that they deal with this by including information from 'Valdes et al. 2019'. I presume they actually mean Armstrong et al. 2019. This basically takes the HADCM3 model and tweaks it by incorporating Dansgaard-Oeschger (DO) and Heinrich events, as seen from ice-core records. This is interesting for the far north of the planet, but is this directly relevant for areas like North Africa and southern Asia? Adding the Armstrong et al. (2019) information on North Atlantic decadal scale variability may be an improvement, but it does still not get to the bottom of the issue which is that precipitation is highly variable in North Africa and the Middle East and presumably was in the past. This variability is fundamental to understanding past human societies in the area: these populations did not live in 'average' conditions, but in conditions that were highly changeable. There is no reason to think that these changes were simply the same as those known from the North Atlantic.

The human humidity niche:

I was quite surprised to see the authors claim that recent humans would not have been able to "greatly change their ecological niche in terms of humidity". I think this is a very dubious claim. What about the invention of water transport technology (perhaps with ostrich eggshell containers sometime ca. 100-60 ka)? What about wells (which as well as the complexities of digging involve a lot of 'politics' about use). What about the use of domestic animals (such as dogs) in recent hunter-gather societies? Such factors (and more could be listed) I think absolutely suggest that it is reasonable to suppose that humans have changed their abilities to survive in arid zones, and give considerable grounds for caution for backfitting contemporary/sub-recent groups to the deep past. I think therefore the Binford data could be used as one end of a model spectrum, a very generous one, but that a much more cautious and realistic one (perhaps 200-300 mm threshold for early humans, reflecting the transition from arid to semi-arid conditions) would be useful.

90 mm a year:

This is very dry! It is classified as 'hyper-arid' in commonly used climate schemes. For comparison, this is the kind of average rainfall seen in areas such as the Empty Quarter of Arabia, the name of which hint at the number of people living in the area even today....and, finally, 90 mm (average) rainfall means very different things in different areas. In Australia, for instance, a thin sand cover means that small water sources are widely spread, this allows humans to live in the area (as long as they are highly mobile). In areas with a thick sand cover, 90 mm a year could mean basically zero available water. So I think just declaring a 90 mm threshold irrespective of the landscape is rather naïve...key point is, how humans survive the best times is not relevant here, nor how they survive the average times (whatever that means), but actually how they survive the worst times. I understand the appeal of the 90 mm 'universal threshold', but thinking how it would actually apply highlights the issues. A snapshot, even a decade, of average rainfall for a vast region is simply not very meaningful. What is meaningful is the climatic conditions conducive to populations spreading across a region, if a one in a hundred year drought killed an entire population, it would be back to square one! I understand that animals like shrews can live in areas with low rainfall, but can humans? Humans are surely obligate drinkers.

We will return to some specifics of the Binford data set, but I think up front saying that early humans lived in hyper-arid conditions is an extreme position. Their position would be stronger if they could cite evidence for early humans living in such arid conditions.

Climate model:

HADCM3 is notorious for under-estimating humidity in the Saharo-Arabian arid belt (i.e. archives showing increased humidity are known from times that the model says it should be dry). They attempt to test their models against a limited number of archives. For precipitation: These are Dead Sea lake levels, southern Red Sea leaf waxes, and XRF humidity India from NW African coast....for temperature, they are O18 from Peqiim and Soreq Caves...there are problems with this. The idea that O18 in central Levantine cave simply reflects temperature change is problematic. It is known that the O18 of these speleothems in large part reflects the 'amount-effect' of the eastern Med, which is prominently driven by Nile input. And then more broadly O18 variation is typically seen as more of an indicator of changes in rainfall rather than temperature. Likewise, I like the idea of trying to tune their model with on the ground indicators, but do this small number achieve that? The Dead Sea is primarily reflecting the winter-rainfall fed central Levant, which reflects Atlantic derived precipitation. The Dead Sea record doesn't look like a particularly good fit to their model to me. For instance, lake level is higher than their model predicts for much of the MIS 3 period (it is around their upper error limit, sometimes in, sometimes out). Lake level was then lower than predicted ca. 130 ka....and higher ca 170-180 and ca 250 ka. So while in part the model captures the broad dynamics, it seems to not work particularly well for the Dead Sea.

So generally, I think they are capturing some pretty basic dynamics, but it is dubious how specifically well it works. The other, much more southerly, sites arguably present a better case, and this is encouraging and interesting. However, the key issue is not how these dynamics change near the equator, but much further north in the Saharo-Arabian arid belt. The records they use do not address this, and this is the key to the question.

They say they compared their precipitation estimates to 'three locations in which paleolakes have been dated to the last interglacial period, following ref 18'. By itself, it does not seem very impressive that their model suggests that some MIS 5 southern Arabian paleolakes (Khujaymah and Saiwan) date to periods of higher rainfall. Once again the issue here is less about the far south of the area, but more about the north. And somewhere like Tayma might indeed provide a key test of the northwards movement of tropical precipitation. However, while the authors claim that 'Lake Tayma' existed 120 ka, no such published lake actually exists (or rather, none has been published)! They are citing for all of this work a paper reviewing paleoclimate models. Perhaps they are referring to lakes in the nearby Nafud Desert. It is true that some have been suggested, but it is a matter of debate what they represent, with some saying they are better characterized as wetlands than lakes (and if this is what they mean, there are also multiple lakes dating to periods such as the later part of MIS 5 when their model suggests things should be dry...the same applies to southern Arabia).

Paleoanthropology:

I would suggest that the rather nebulous term 'anatomically modern humans' is avoided. They should just talk about *Homo sapiens*. They cite Stringer on this, but in the paper cited, he does not use the term AMH. They should cite more recent work on the origin and evolution of *Homo sapiens*, such as Scerri and colleagues (2018). The concept of 'AMH' has a lot of conceptual baggage attached which is simply not needed in this study....and there are various small points which could be mentioned. Easier just to talk about *Homo sapiens*. Or perhaps just 'modern humans' if they are trying to make some kind of point...Other smaller points could be raised, why cite a paper on dating Skhul from the 1980's and not more recent papers (e.g. Grün et al., 2005)?....the recent findings from Misliya and Apidima both rely on U-series dates which may be problematic (see for instance: DOI: 10.1126/science.aat6598). I would suggest they caveat more and say there are possible preliminary hints of Middle Pleistocene *Homo sapiens* in Eurasia, but they need more work to be confirmed. I think this is important as the key question of how many dispersals were there out of Africa is currently very unclear; so the various claims for *Homo sapiens* at different times should not be combined with the

southern route climate presented in this paper to suggest that there were necessarily repeated periods when *Homo sapiens* could (or rather, did) leave Africa.

Red Sea crossing:

Given that their northern route conclusion basically mirrors current views (although they bring a valuable quantification), their real innovation is the suggestion of an extended window of dispersal by the southern route ca. 60-40 ka. As well as being rather incongruous with regional paleoclimate (see below), this presupposes a Red Sea crossing. While they do add some caveats, I think these appear as rather post hoc additions, and are still too charitable. For instance, they claim that the site of Abdur Reef provides evidence of maritime "adaptations", yet it appears to simply be a natural death assemblage of oysters etc, mixed together with some redeposited stone tools and terrestrial mammal remains. The site provides evidence that early humans sometimes lived near the sea: that is it. So I think these issues could be dealt with more effectively. Perhaps by emphasizing in earlier part of the paper their reconstruction of regional scale paleoclimate, and then in the later part of the paper exploring the implications of this in terms of real-world application. Of course it is possible that the Red Sea was crossed, but it should very clearly be pointed out that there is no evidence from within thousands of kilometers and tens of thousands of years for early sites which would support this. Likewise, they could consider in the later part of the paper why the archaeological record of southern Arabia is so incongruous with their narrative. SD-1 at 55 ka and assemblages B (between ca. 120 and 40 ka) and A (ca 40 ka) at Jebel Faya are all dominated by unique localized characteristics (Armitage et al., 2011 *Science*; Delagnes et al., 2012 *JHE*). Is this not a bit strange if there was a huge easy dispersal window across southern Arabia? The expectations of repeated easy dispersals across Red Sea are clear: yet despite a lot of looking, the kind of archaeological material which would indicate recent arrivals from East Africa appears to be lacking.

Regional paleoclimate:

I think more thought needs to be given to comparing their results to regional paleoclimate archives. They do some of this, and some aspects are useful (the pollen comparisons, for instance, look strong). In other regards, however, their results are less convincing. I think they could see this in two regards 1) tuning their model (which they do relatively well in some regards) and 2) more thought on why their results are different from the results of previous studies....At present the latter is lacking. Aside from the issues of crossing the Red Sea, their climate reconstruction is at odds with regional paleoclimate as known from dozens of papers. For instance, the speleothem record (see most recently Nicholson et al., 2020) suggests aridity for the tens of millennia of apparently high rainfall in the mid-Late Pleistocene. Maybe this is because rainfall is below a threshold for speleothem formation, maybe so, but this at least needs to be discussed. The paleoclimate of Arabia is fairly well resolved, and the authors of this paper are suggesting a very different view and basically ignoring previous work. The fluvio-lacustrine from Wadi Mistal, for instance, offers some useful perspectives (Hoffman et al., 2015 *QI*). Here in early MIS 3, 112 fining-up sequences represent major storm events. That is the form that increased rainfall took in the area. So are average annual or decadal values that pertinent? What impact would extreme aridity between these storms have had on human societies?

Binford dataset:

I think the key issues with this paper come from a simplistic lifting of the Binford dataset and its application into the study. There are notorious issues with using Binford's compiled data in this way. For instance, many of the populations in his sample used metal tools, have domestic animals (such as dogs) etc etc. This alone makes them dubious as analogues for the ancient past. The Binford dataset is highly spatially biased, consisting overwhelmingly of North American and Australian groups. The

authors acknowledge that, but think that there is still likely to be an overall ecological level pattern here, their '90mm rule'. I am dubious of that. In most regions, the hunter-gatherers were as Binford puts it (p 132), "integrated into non-hunter-gatherer systems at the time of description". These are simply not unchanged groups which can represent the ancient past. Binford is very clear "hunter-gatherers are not found in true desert settings, which are the most marginal habitats in terms of plant productivity. They are rarely found in zones of semidesert scrub" (p 136)...this semi-desert scrub is defined in his system as having average rainfall of 242 mm (p 97)....The Binford dataset is a valuable resource, no doubt, but I think that the authors make too extreme a use of it (given the various issues discussed above) and this leads to the naïve '90 mm rule'.

Conclusion:

In summary, I like a lot about this paper, and the basic logic is absolutely correct. However, I feel the factors discussed above are problematic. I would suggest that a stronger paper would come from a more bounded approach, where they use the '90mm rule' they have used here with a more realistic threshold (say 200-300 mm?). This would give an interesting upper and lower bounds, and I think make the paper a lot more useful than with the present rather dubious 90 mm use. It would be very interesting to see the results of this more cautious (/realistic) level of precipitation tolerance.

And then they should be more explicit about the meaning of the results. They give regional scale insights, not 'corridors'. How these climate conditions actually played out in the real world then depended on a variety of factors from physical geography to aspects of human demography and behavior which are currently treated in a rather simplistic way.

Reviewer #2:

Remarks to the Author:

This is the second time that I review this manuscript and I am pleased to see that the recommendations that I made previously have been taken into account in this new and definitely improved version. I reiterate that this is a well written manuscript that addresses some important concepts in human evolution and ecology that will be of interest for a large audience. This manuscript is well executed and tackled hypotheses of human expansion out of Africa via robust modelling and quantitative analyses. I only have few recommendations left (mostly to improve the discussion).

I do appreciate the time and effort that the authors put to come up with this revised version and once my points will be addressed, I will be happy to recommend this manuscript for publication in Nature Communications.

Main comments:

The temporal resolution: since the exit windows are calculated at a decadal scale, I wonder whether there could be a lag effect between the opening of the exit pathway and that time human would need to make their move. In other word, a discussion about how long an exit window should be for human to seize the opportunity to move accounting for demographic processes and adaptative behaviour (I doubt that it would be an immediate response)?

The notions of "demography" and "migration" are vaguely mentioned in the discussion and in general largely overlooked. I do think that the discussion would greatly benefit from a better treatment of those notions. I am aware that it might seem to be out of scope but discussing these processes

against your results would broaden the context of your study. It should be mentioned for example:

- demography: (i) what would be the minimal population size required to manage to cross those new pathways. (ii) Please expand on the "demographic rescue" process. (iii) How the change in environment would influence human demography, not only in term of survival but in term of fertility, density dependence (density feedback as a function of landscape carrying capacity, etc...)

- migration: how fast could AMH travel? Would AHM travel as fast across an unknown territory as in a familiar environment? The migration speed across those corridors would tightly be related to the demography (e.g., one or two person moving vs a big group of people) and human behaviour (e.g., moving randomly or driven by specific landscape feature).

Minor comments:

L 68-69: could you be more specific about what you mean by "humidity constraint"? Do you mean "drinking water availability? If so, what is the logic behind that? If AMH have the technology to cross seas, we can safely assume that they are also capable to carry drinking water, right? This assumption needs to be justified a bit better.

L 94-95: please define what you mean by 'short term palaeoclimatic variability'?

There are some inconsistencies in the notation of years e.g., 'k', 'k years ago' or 'kya'

L 128: please clarify "suitably wet climate". Do you mean in term of volume of rainfall (high) and aridity (low)?

Regards

Dr Frédéric Saltré

Reviewer #1 (Remarks to the Author):

I have mixed feelings about this paper. In positive terms, it is exactly the kind of study that is needed, i.e. quantitative reconstruction of regional paleoenvironmental conditions in the context of debates about the timing and character of dispersals out of Africa. However, I think that there are significant problems with some of the specifics of the manuscript at it stands. The study has the potential be highly novel and interesting to researchers in multiple disciplines. I think if the analyses can be repeated with a more realistic water-tolerance level then this will be a highly significant and influential paper.

The authors response to reviewers on the previous version of the paper appear generally reasonable, yet they are also somewhat cosmetic and the substantial methods and conclusions of the paper remain the same. But by doing things like adding an aridity index, they have improved the paper...I just think that they have not gone far enough with the changes. Likewise, adding a 5% error buffer is good, but likewise I don't think it's enough.

As described in detail further below, we now present windows of climatic connectivity between Eurasia and Africa based on a continuous range of precipitation tolerance thresholds between 60 mm y^{-1} and 200 mm y^{-1} , and have added a discussion of the implications of thresholds different to our baseline estimate of 90 mm y^{-1} on the resulting windows.

Aims and methods: If the aim is looking for a dispersal 'corridor', things could be much easier: the Nile Valley offers a corridor for much of the relevant time period. Plus given the importance of physical geography factors (e.g. a sea crossing being involved in the southern route) mean that talk of 'corridors' is rather illusory. I think the aim of the paper should be cast more as about reconstructing regional environmental conditions – an important goal, given that most current paleoenvironmental archives offer scattered and often poorly dated insights. Their model arguably tells us more about what we could call 'survivability' than 'dispersability'. How conditions in a particular area might relate to dispersal routes needs, in my opinion, rather more nuanced consideration.

The key specific point here is about whether early humans crossed the southern Red Sea. Crossing a sea with strong currents is a strange 'corridor' for a terrestrially adapted animal (which does not mean it was not used, but just it seems a rather strange 'corridor'). If we see the dispersal of human populations across thousands of kilometers of challenging landscapes as an 'event' (as in the 'out of Africa event' that they, following common terminology in genetics, use), then describing a 'corridor' makes sense, but in the applied context of ecology, environment and human social dynamics it is a rather problematic concept. While I think the concepts of corridors as applied here is dubious, the one sense in which it is perhaps useful – that is river systems which cross the arid areas (e.g. Drake et al., 2011 in PNAS) – is actually ignored here. I understand why that is so, but they should then be more explicit about what they are doing being about reconstructing regional scale patterns, not actual corridors.

We understand that the use of the term 'corridor' may have been misleading. We have removed it from the text and have rephrased the relevant sections. We have addressed the crossing the Red Sea in more depth as described in our response to the Reviewer's specific query on this further below.

Survivability and dispersibility: One of the things the authors have changed in this new version of the paper is an attempt to more explicitly deal with the highly variable nature of climate in these areas. In my opinion this is absolutely fundamental. It doesn't really matter if ten years have high enough rainfall for survival if there is not a drop in year eleven. The authors suggest in the response to reviewers that they deal with this by including information from 'Valdes et al. 2019'. I presume they actually mean Armstrong et al. 2019. This basically takes the HADCM3 model and tweaks it by incorporating Dansgaard-Oeschger (DO) and Heinrich events, as seen from ice-core records. This is

interesting for the far north of the planet, but is this directly relevant for areas like North Africa and southern Asia? Adding the Armstrong et al. (2019) information on North Atlantic decadal scale variability may be an improvement, but it does still not get to the bottom of the issue which is that precipitation is

highly variable in North Africa and the Middle East and presumably was in the past. This variability is fundamental to understanding past human societies in the area: these populations did not live in 'average' conditions, but in conditions that were highly changeable. There is no reason to think that these changes were simply the same as those known from the North Atlantic.

The Reviewer is correct that the relevant reference is Armstrong et al. 2019, and we apologise for this mistake in our rebuttal. Our main interest in using the dataset presented in this paper is not specifically the simulation of Dansgaard-Oeschger and Heinrich events, but the high temporal resolution of the data, which had allowed us to consider decadal scale variability within a given millennium. We would not agree that the design of HadCM3 is such that the treatment of Dansgaard-Oeschger and Heinrich events by Armstrong et al. would result in the simulated climatic variability in North Africa and the Middle East being the same as in the North Atlantic. We fully agree with the Reviewer about the relevance of short-term climatic variability for out-of-Africa migrations, and we have added the following paragraph to highlight this issue:

Our analysis of the climatic feasibility of *Homo sapiens* leaving the African continent in the last 300k years is based on decadal scale variability of annual precipitation and aridity. Whilst neither empirical nor simulation-based approaches currently appear capable of reconstructing climatic conditions at a higher temporal resolution across the same time period and geographical area without compromising robustness, it is important to bear in mind that short-term climatic variability can play an important role for human population dynamics. Storms and monsoon rains followed by extended dry spells would have posed different challenges than the same total rainfall spread out over a long period.

The human humidity niche: I was quite surprised to see the authors claim that recent humans would not have been able to "greatly change their ecological niche in terms of humidity". I think this is a very dubious claim. What about the invention of water transport technology (perhaps with ostrich eggshell containers sometime ca. 100-60 ka)? What about wells (which as well as the complexities of digging involve a lot of 'politics' about use). What about the use of domestic animals (such as dogs) in recent hunter-gather societies? Such factors (and more could be listed) I think absolutely suggest that it is reasonable to suppose that humans have changed their abilities to survive in arid zones, and give considerable grounds for caution for backfitting contemporary/sub-recent groups to the deep past. I think therefore the Binford data could be used as one end of a model spectrum, a very generous one, but that a much more cautious and realistic one (perhaps 200-300 mm threshold for early humans, reflecting the transition from arid to semi-arid conditions) would be useful.

We have removed the sentence pointed out by the Reviewer on the change of the human humidity niche over time. Following this, and the Reviewer's comments on the use of the Binford dataset below, we have moved the relevant paragraph from the Methods to the main text Discussion, and have pointed out additional limitations noted by the Reviewer, including a potentially less developed ability to store and transport water (see our response further below).

90 mm a year: This is very dry! It is classified as 'hyper-arid' in commonly used climate schemes. For comparison, this is the kind of average rainfall seen in areas such as the Empty Quarter of Arabia, the name of which hint at the number of people living in the area even today....and, finally, 90 mm (average) rainfall means very different things in different areas. In Australia, for instance, a thin sand

cover means that small water sources are widely spread, this allows humans to live in the area (as long as they are highly mobile). In areas with a thick sand cover, 90 mm a year could mean basically zero available water. So I think just declaring a 90 mm threshold irrespective of the landscape is rather naïve...key point is, how humans survive the best times is not relevant here, nor how they survive the average times (whatever that means), but actually how they survive the worst times. I understand the appeal of the 90 mm 'universal threshold', but thinking how it would actually apply highlights the issues. A snapshot, even a decade, of average rainfall for a vast region is simply not very meaningful. What is meaningful is the climatic conditions conducive to populations spreading across a region, if a one in a hundred year drought killed an entire population, it would be back to square one! I understand that animals like shrews can live in areas with low rainfall, but can humans? Humans are surely obligate drinkers.

We will return to some specifics of the Binford data set, but I think up front saying that early humans lived in hyper-arid conditions is an extreme position. Their position would be stronger if they could cite evidence for early humans living in such arid conditions.

To accommodate the Reviewer's comment, we have restructured our manuscript in that we now first present the estimated minimum thresholds required to exit Africa in the past 300k years, which are derived only from the climate reconstructions (Fig. 1). We then go on to use these data to infer climatic windows of opportunity assuming precipitation tolerance thresholds covering a continuous range of values from 60 mm y^{-1} to 200 mm y^{-1} (Fig. 2). Whilst we continue to use our previously derived thresholds (90 mm y^{-1} for precipitation, 1.7 for Köppen aridity) as baselines in the discussion of our results, we use our analysis in Fig. 2 to examine the implications of higher thresholds for the resulting windows in the following newly added paragraph:

Assuming that precipitation and aridity tolerance thresholds derived from contemporary hunter-gatherer data can be used as proxies of those of early modern humans is not without limitations. Ethnographically documented populations are not uniformly distributed across the world, residing predominantly in North America and Australia, South America, Sub-Saharan Africa, and South and Southeast Asia ³², where both climatic and soil hydrological conditions can differ significantly from those in Northern Africa and Southwest Asia. In addition, technological differences, such as in the ability to dig wells and store and transport water, may imply higher threshold levels for early hunter-gatherers. Whilst currently available evidence may not allow us to quantitatively refine the tolerance thresholds derived from contemporary data, we can investigate the effect of different thresholds on the resulting windows of climatic connectivity between Africa and Eurasia (Fig. 2b,c). Along the northern route, our data suggest that precipitation tolerance thresholds above 110 mm y^{-1} would have likely allowed for expansions out of Africa within the previously estimated windows only during shorter intervals with rainfall levels above the millennium-scale average. In this scenario, conditions would have been most favourable during the last interglacial period 130k years ago. For tolerance thresholds above 130 mm y^{-1} , migration would have likely been very challenging and restricted to unusually wet intervals. The southern route would have provided more scope for lower precipitation tolerance levels. Thresholds up to 200 mm y^{-1} would have provided opportunities to leave Africa during the last interglacial period. Between this point and the wet Holocene, tolerance levels of up to 130 mm y^{-1} would likely have allowed for migrations into Eurasia during favourable intervals between 65k and 55 years ago.

Climate model: HADCM3 is notorious for under-estimating humidity in the Saharo-Arabian arid belt (i.e. archives showing increased humidity are known from times that the model says it should be dry). They attempt to test their models against a limited number of archives. For precipitation: These

are Dead Sea lake levels, southern Red Sea leaf waxes, and XRF humidity India from NW African coast....for temperature, they are O18 from Peqim and Soreq Caves...there are problems with this. The idea that O18 in central Levantine cave simply reflects temperature change is problematic. It is known that the O18 of these speleothems in large part reflects the 'amount-effect' of the eastern Med, which is prominently driven by Nile input. And then more broadly O18 variation is typically seen as more of an indicator of changes in rainfall rather than temperature. Likewise, I like the idea of trying to tune their model with on the ground indicators, but do this small number achieve that? The Dead Sea is primarily reflecting the winter-rainfall fed central Levant, which reflects Atlantic derived precipitation. The Dead Sea record doesn't look like a particularly good fit to their model to me. For instance, lake level is higher than their model predicts for much of the MIS 3 period (it is around their upper error limit, sometimes in, sometimes out). Lake level was then lower than predicted ca. 130 ka....and higher ca 170-180 and ca 250 ka. So while in part the model captures the broad dynamics, it seems to not work particularly well for the Dead Sea.

So generally, I think they are capturing some pretty basic dynamics, but it is dubious how specifically well it works. The other, much more southerly, sites arguably present a better case, and this is encouraging and interesting. However, the key issue is not how these dynamics change near the equator, but much further north in the Saharo-Arabian arid belt. The records they use do not address this, and this is the key to the question.

They say they compared their precipitation estimates to 'three locations in which paleolakes have been dated to the last interglacial period, following ref 18'. By itself, it does not seem very impressive that their model suggests that some MIS 5 southern Arabian paleolakes (Khujaymah and Saiwan) date to periods of higher rainfall. Once again the issue here is less about the far south of the area, but more about the north. And somewhere like Tayma might indeed provide a key test of the northwards movement of tropical precipitation. However, while the authors claim that 'Lake Tayma' existed 120 ka, no such published lake actually exists (or rather, none has been published)! They are citing for all of this work a paper reviewing paleoclimate models. Perhaps they are referring to lakes in the nearby Nafud Desert. It is true that some have been suggested, but it is a matter of debate what they represent, with some saying they are better characterized as wetlands than lakes (and if this is what they mean, there are also multiple lakes dating to periods such as the later part of MIS 5 when their model suggests things should be dry...the same applies to southern Arabia).

We appreciate the Reviewer's interest in a thorough validation of the simulated data by means of empirical records. We would reiterate that suitable long-term quantitative proxy data from the region in question are very sparse and not without limitations. We acknowledge that the empirical data used in the validation do not cover all parts of Northern Africa and the Arabian peninsula; however, the lack of appropriate data poses a fundamental barrier to providing better spatial coverage. As a point of reference, the climate model validation in the *global* study of Timmermann and Friedrich (2016, Nature) on climatic drivers of human dispersal contains only one precipitation-related long-term proxy; plus, almost all temperature-related proxies are marine or arctic. Similarly, the validation of the precipitation data simulated by Armstrong et al. (2019, Nature Scientific Data) for the entire Northern hemisphere consists of only one proxy from Greenland.

We feel that we have appropriately pointed out the issue that the long-term proxy records in Fig. 4 are also partially affected by environmental variables other than precipitation. In our view, it is not possible to filter these factors out, leaving us with the choice between removing these proxy comparisons altogether or keeping them as a way to illustrate that the modelled data correctly reproduce major qualitative cycles. We feel that despite the limitations, and given the lack of better alternative data, the manuscript benefits from the comparisons.

Regarding two specific comments of the Reviewer: As delta18O changes directly as the result of temperature fluctuations, it can, and has previously, been used for the validation of long-term temperature dynamics. We agree with the Reviewer that it is also affected by other climatic variables, and we have further emphasised this in the text; however, as in the case

of precipitation, the lack of better proxies suitable for validating simulated long-term terrestrial temperature dynamics limits our choice of empirical data. We feel again that, despite certain limitations, the comparison of temperature against long-term empirical proxies provides a valuable addition to the additional comparison against pollen-based reconstructions, which are available only for two relatively recent points in time. We have also changed the name 'Lake Tayma' to 'Nefud' in Supplementary Figure 4 to avoid confusion.

Paleoanthropology: I would suggest that the rather nebulous term 'anatomically modern humans' is avoided. They should just talk about *Homo sapiens*. They cite Stringer on this, but in the paper cited, he does not use the term AMH. They should cite more recent work on the origin and evolution of *Homo sapiens*, such as Scerri and colleagues (2018). The concept of 'AMH' has a lot of conceptual baggage attached which is simply not needed in this study....and there are various small points which could be mentioned. Easier just to talk about *Homo sapiens*. Or perhaps just 'modern humans' if they are trying to make some kind of point...Other smaller points could be raised, why cite a paper on dating Skhul from the 1980's and not more recent papers (e.g. Grün et al., 2005)?....the recent findings from Misliya and Apidima both rely on U-series dates which may be problematic (see for instance: DOI: 10.1126/science.aat6598). I would suggest they caveat more and say there are possible preliminary hints of Middle Pleistocene *Homo sapiens* in Eurasia, but they need more work to be confirmed. I think this is important as the key question of how many dispersals were there out of Africa is currently very unclear; so the various claims for *Homo sapiens* at different times should not be combined with the southern route climate presented in this paper to suggest that there were necessarily repeated periods when *Homo sapiens* could (or rather, did) leave Africa.

Following the Reviewer's suggestion, we have removed all instances of "AMHs" from the text and now use "*Homo sapiens*" where appropriate.

We thank the Reviewer for their suggestion of an updated reference on dating Skhul, which we have added to the text as recommended.

We have added the reference questioning the estimated timing for *Homo Sapiens* in Israel, pointed out by the Reviewer, to the text, and have modified the wording in the Introduction to emphasise more clearly the uncertainty in the existing evidence for Middle Pleistocene *Homo sapiens* in Eurasia as well as the need for further confirmatory work.

Red Sea crossing: Given that their northern route conclusion basically mirrors current views (although they bring a valuable quantification), their real innovation is the suggestion of an extended window of dispersal by the southern route ca. 60-40 ka. As well as being rather incongruous with regional paleoclimate (see below), this presupposes a Red Sea crossing. While they do add some caveats, I think these appear as rather post hoc additions, and are still too charitable. For instance, they claim that the site of Abdur Reef provides evidence of maritime "adaptations", yet it appears to simply be a natural death assemblage of oysters etc, mixed together with some redeposited stone tools and terrestrial mammal remains. The site provides evidence that early humans sometimes lived near the sea: that is it. So I think these issues could be dealt with more effectively. Perhaps by emphasizing in earlier part of the paper their reconstruction of regional scale paleoclimate, and then in the later part

of the paper exploring the implications of this in terms of real-world application. Of course it is possible that the Red Sea was crossed, but it should very clearly be pointed out that there is no evidence from within thousands of kilometers and tens of thousands of years for early sites which would support this. Likewise, they could consider in the later part of the paper why the archaeological record of southern Arabia is so incongruous with their narrative. SD-1 at 55 ka and assemblages B (between ca. 120 and 40 ka) and A (ca 40 ka) at Jebel Faya are all dominated by unique localized characteristics (Armitage et al., 2011 Science; Delagnes et al., 2012 JHE). Is this not a bit strange if there was a huge easy dispersal window across southern Arabia? The expectations of

repeated easy dispersals across Red Sea are clear: yet despite a lot of looking, the kind of archaeological material which would indicate recent arrivals from East Africa appears to be lacking.

Following the Reviewer's suggestion, we have restructured the text in that we now begin by examining possible dispersal windows purely from a climatic perspective, and only later discuss our results for the Southern route in the context of whether or not it was possible to cross the Red Sea. We have removed the elements pointed out as potentially controversial by the Reviewer in our previous version, and replaced our discussion of this topic by with the following paragraph:

In addition to climatic constraints, the requirement to cross the strait of the Bab e-Mandeb would have posed a key challenge to migration along the southern route. Whether early hunter-gatherers ventured across the Red Sea, as has been suggested based notably on genetic evidence^{40,41}, remains a subject of debate, given the very limited archaeological support for this scenario⁴². During periods of low sea level, the Arabian peninsula was likely visible from present-day Djibouti and southeast Eritrea, and crossing the strait at these times may not have required sophisticated boats or seafaring skills⁴³. However, whilst it is likely that humans lived on the western Red Sea coast and used marine food resources⁴⁴, direct evidence of boats and maritime travel remains to be found. In addition, although technological similarities have been suggested between some sites on the Arabian Peninsula and in Northeast Africa⁴⁵, other sites show no such relationship⁴⁶. Caution should therefore be taken before interpreting the more favourable climate along the southern route suggested by our data as evidence for a southern exit; instead, expanding and reconciling genetic and archaeological lines of evidence remains crucial for clarifying the role of the Bab e-Mandeb strait in expansions out of Africa.

Whilst we very much aim to provide a nuanced view of the different arguments on whether maritime travel across the Red Sea was possible, we feel that given the ongoing nature of the debate and the diverging interpretations of archaeological and genetic evidence, it would not be within the scope of our paper to take a strong position in favour of either scenario.

Regional paleoclimate: I think more thought needs to be given to comparing their results to regional paleoclimate archives. They do some of this, and some aspects are useful (the pollen comparisons, for instance, look strong). In other regards, however, their results are less convincing. I think they could see this in two regards 1) tuning their model (which they do relatively well in some regards) and 2) more thought on why their results are different from the results of previous studies...At present the latter is lacking. Aside from the issues of crossing the Red Sea, their climate reconstruction is at odds with regional paleoclimate as known from dozens of papers. For instance, the speleothem record (see most recently Nicholson et al., 2020) suggests aridity for the tens of millennia of apparently high rainfall in the mid-Late Pleistocene. Maybe this is because rainfall is below a threshold for speleothem formation, maybe so, but this at least needs to be discussed. The paleoclimate of Arabia is fairly well resolved, and the authors of this paper are suggesting a very different view and basically ignoring previous work. The fluvio-lacustrine from Wadi Mistal, for instance, offers some useful perspectives (Hoffman et al., 2015 QI). Here in early MIS 3, 112 fining-up sequences represent major storm events. That is the form that increased rainfall took in the area. So are average annual or decadal values that pertinent? What impact would extreme aridity between these storms have had on human societies?

From the references mentioned by the Reviewer, and our own literature review on Arabian palaeoenvironmental conditions, we were not able to find quantitative data that clearly contradict our reconstructions. We would reiterate that areas estimated to have been inhabitable in our analysis may indeed have been arid (and identified as such in empirical

studies) and far from characterised by lush vegetation. Speleotherms provide a well dated record compared to other proxies such as alluvial fans, but there is a major issue on how to interpret them, as they only form when precipitation is above a certain threshold (which can be location specific). Low precipitation levels are unlikely to lead to speleotherm formation, so the climatic regimes discussed in our manuscripts are likely to be below the resolution of speleotherms.

We have added the following paragraph to address the issue of seasonal variability highlighted by the Reviewer:

Our analysis of the climatic feasibility of Homo sapiens leaving the African continent in the last 300k years is based on decadal scale variability of annual precipitation and aridity. Whilst neither empirical nor simulation-based approaches currently appear capable of reconstructing climatic conditions at a higher temporal resolution across the same time period and geographical area without compromising robustness, it is important to bear in mind that short-term climatic variability can play an important role for human population dynamics. Storms and monsoon rains followed by extended dry spells would have likely been more challenging for early humans to tolerate than the same total rainfall spread out over a long period.

Binford dataset: I think the key issues with this paper come from a simplistic lifting of the Binford dataset and its application into the study. There are notorious issues with using Binford's compiled data in this way. For instance, many of the populations in his sample used metal tools, have domestic animals (such as dogs) etc etc. This alone makes them dubious as analogues for the ancient past. The Binford dataset is highly spatially biased, consisting overwhelmingly of North American and Australian groups. The authors acknowledge that, but think that there is still likely to be an overall ecological level pattern here, their '90mm rule'. I am dubious of that. In most regions, the hunter-gatherers were as Binford puts it (p 132), "integrated into non-hunter-gatherer systems at the time of description". These are simply not unchanged groups which can represent the ancient past. Binford is very clear "hunter-gatherers are not found in true desert settings, which are the most marginal habitats in terms of plant productivity. They are rarely found in zones of semidesert scrub" (p 136)...this semi-desert scrub is defined in his system as having average rainfall of 242 mm (p 97)...The Binford dataset is a valuable resource, no doubt, but I think that the authors make too extreme a use of it (given the various issues discussed above) and this leads to the naïve '90 mm rule'.

We have added the following paragraph to point out more clearly the limitations of the Binford data:

Assuming that precipitation and aridity tolerance thresholds derived from contemporary hunter-gatherer data can be used as proxies of those of early modern humans is not without limitations. Ethnographically documented populations are not uniformly distributed across the world, residing predominantly in North America and Australia, South America, Sub-Saharan Africa, and South and Southeast Asia ³², where both climatic and soil hydrological conditions can differ significantly from those in Northern Africa and Southwest Asia. In addition, technological differences, such as in the ability to store and transport water ⁴⁰, may imply higher threshold levels for early hunter-gatherers. Whilst currently available evidence may not allow us to quantitatively refine the tolerance thresholds derived from contemporary data, we can investigate the effect of different thresholds on the resulting windows of climatic connectivity between Africa and Eurasia (Fig. 2b,c). [...]

Conclusion: In summary, I like a lot about this paper, and the basic logic is absolutely correct. However, I feel the factors discussed above are problematic. I would suggest that a stronger paper

would come from a more bounded approach, where they use the '90mm rule' they have used here with a more realistic threshold (say 200-300 mm?). This would give an interesting upper and lower bounds, and I think make the paper a lot more useful than with the present rather dubious 90 mm use. It would be very interesting to see the results of this more cautious (/realistic) level of precipitation tolerance.

And then they should be more explicit about the meaning of the results. They give regional scale insights, not 'corridors'. How these climate conditions actually played out in the real world then depended on a variety of factors from physical geography to aspects of human demography and behavior which are currently treated in a rather simplistic way.

As described above, we have added a new figure visualising windows of climatic connectivity between Eurasia and Africa for a range of precipitation tolerance thresholds, and have added a discussion of the impact of different thresholds on the resulting windows.

We have also added the following paragraph to clarify the scope of our analysis, specifically in regard to human demography, as pointed out by the Reviewer:

Our results provide estimates of when migration between Africa and Eurasia would have been climatically feasible for *Homo sapiens*, not whether these potential windows of opportunity were seized. This would require combining our data with human dispersal models that explicitly simulate spatio-temporal population dynamics, which we have not attempted here. Uncertainties associated with such an approach would likely be substantial given that current estimates of parameters relevant to demographic processes, such as population growth rates and dispersal speeds, range across several orders of magnitude^{23,24}. Likewise, important questions such as to what extent movement patterns of early modern humans were directional or random, and how they varied in different environments and in response to changes in population size, largely lack quantitative answers. Consolidating anthropological, archaeological, and genetic data would seem as the most promising avenue towards reducing existing uncertainties.

Reviewer #2 (Remarks to the Author):

This is the second time that I review this manuscript and I am pleased to see that the recommendations that I made previously have been taken into account in this new and definitely improved version. I reiterate that this is a well written manuscript that addresses some important concepts in human evolution and ecology that will be of interest for a large audience. This manuscript is well executed and tackled hypotheses of human expansion out of Africa via robust modelling and quantitative analyses. I only have few recommendations left (mostly to improve the discussion).

I do appreciate the time and effort that the authors put to come up with this revised version and once my points will be addressed, I will be happy to recommend this manuscript for publication in Nature Communications.

The temporal resolution: since the exit windows are calculated at a decadal scale, I wonder whether there could be a lag effect between the opening of the exit pathway and that time human would need to make their move. In other word, a discussion about how long an exit window should be for human to seize the opportunity to move accounting for demographic processes and adaptative behaviour (I doubt that it would be an immediate response)?

The notions of "demography" and "migration" are vaguely mentioned in the discussion and in general largely overlooked. I do think that the discussion would greatly benefit from a better treatment of those notions. I am aware that it might seems to be out of scope but discussing these

processes against your results would broaden the context of your study. It should be mentioned for example:

- demography: (i) what would be the minimal population size required to manage to cross those new pathways. (ii) Please expand on the “demographic rescue” process. (iii) How the change in environment would influence human demography, not only in term of survival but in term of fertility, density dependence (density feedback as a function of landscape carrying capacity, etc...)
- migration: how fast could AMH travel? Would AHM travel as fast across an unknown territory as in a familiar environment? The migration speed across those corridors would tightly be related to the demography (e.g., one or two person moving vs a big group of people) and human behaviour (e.g., moving randomly or driven by specific landscape feature).

We fully agree with the Reviewer that understanding the demographic dynamics of early modern humans is important for improving our reconstructions of past migration fluxes out of Africa. Given the substantial uncertainty currently associated with demographic parameters, we have not attempted to explicitly simulate spatio-temporal demographics. We have added the following paragraph to discuss this aspect:

Our results provide estimates of when migration between Africa and Eurasia would have been climatically feasible for *Homo sapiens*, not whether these potential windows of opportunity were seized. This would require combining our data with human dispersal models that explicitly simulate spatio-temporal population dynamics, which we have not attempted here. Uncertainties associated with such an approach would likely be substantial given that current estimates of parameters relevant to demographic processes, such as population growth rates and dispersal speeds, range across several orders of magnitude^{23,24}. Likewise, important questions such as to what extent movement patterns of early modern humans were directional or random, and how they varied in different environments and in response to changes in population size, largely lack quantitative answers. Consolidating anthropological, archaeological, and genetic data would seem as the most promising avenue towards reducing present uncertainties.

L 68-69: could you be more specific about what you mean by “humidity constraint”? Do you mean “drinking water availability? If so, what is the logic behind that? If AMH have the technology to cross seas, we can safely assume that they are also capable to carry drinking water, right? This assumption needs to be justified a bit better.

We have added the following phrase to the sentence on why we focus on precipitation/ardity:

“[...] climatic constraints of the ability of early modern humans to sustain themselves, by hunting and gathering animal- and plant-based food and collecting drinking water²⁸ [...]”,

Ref. 28 provides time estimates of the use of different types of receptacles for storing drinking water, the first one dating back to before the appearance of *Homo sapiens*.

L 94-95: please define what you mean by ‘short term palaeoclimatic variability’?

We have replaced “short-term” with “annual”.

There are some inconsistencies in the notation of years e.g., ‘k’, ‘k years ago’ or ‘kya’

We have removed all instances of kya, and now use the “xk years ago” or “between xk and yk years ago” throughout the manuscript.

L 128: please clarify “suitably wet climate”. Do you mean in term of volume of rainfall (high) and aridity (low)?

We have replaced this phrase with “sufficient rainfall”.

Reviewers' Comments:

Reviewer #1:

Remarks to the Author:

I am very pleased to see the way that the authors have revised this paper. As I stated in my previous review, I was very supportive of the kind of approach used in the paper, but I had multiple specific issues with particular aspects. I think that the authors have adequately responded to the reviews. The revised paper is much improved and I think it is ready to be published.

I do not necessarily agree with the authors suggestion about a 90 mm threshold, but they have defended their position and I am satisfied by the way they caveat their results and interpretations. Personally I think it is unlikely that many dispersals were happening when rainfall was below 200 or more mm – but the point is that their data and figures etc can be used in discussing such a position. In other words, there is a clear distinction between the data and their interpretation of it. This is how it should be with all papers, but all too often in such topics there is no separation of description and interpretation.

One very minor thing that I would suggest is that given the authors argue for an “an unprecedentedly long period of largely favourable climate between ~65k and ~30k years ago” in southern Arabia, they should mention that despite decades of study, no evidence of this has been found. There is evidence for minor wet phases such as ca 55 ka, but not for a ca. 35 thousand year wet phase that is a major conclusion of their study. This is not to say that this wet phase did not happen – but climate archives showing it have not been found. As suggested in my previous review, I think the way they have calibrated their climate model against possibly inappropriate archives (e.g. dubious to understand monsoonal dynamics using the Dead Sea record) may be the reason for an anomalously prolonged southern Arabian wet phase ca. 65 to 35 ka. I would appreciate if they mentioned an absence of evidence for a long southern Arabian humid period in the proposed time. They could mention that testing such a model should be a priority for future palaeoclimate research in the region. An alternative interpretation of the major dispersal ca. 70-50 ka is that a small population which had passed through the arid belt expanded, spread and split; they may have already been in more humid regions. So the apparent correlation between the ‘successful’ dispersal period and the hypothesized 65-30 ka major southern Arabian wet phase is tempting, but I would appreciate if some caveats with briefly mentioned.

Minor points.

‘.’ missing in email address for corresponding author on main text file.

They say Arabia was ‘likely visible’ from Africa. It absolutely would have been visible.

“competing with Neanderthal” should be Neanderthals.

“Denisova” should be Denisovans.

I don’t understand what the ‘Nefud’ lake is in supplementary figure 4. And no reference is provided. There are many lakes in the Nefud, of lots of different ages.

In summary, I have these few minor suggestions for edits, otherwise I think the paper is ready and should be published.

Reviewer #1:

I am very pleased to see the way that the authors have revised this paper. As I stated in my previous review, I was very supportive of the kind of approach used in the paper, but I had multiple specific issues with particular aspects. I think that the authors have adequately responded to the reviews. The revised paper is much improved and I think it is ready to be published.

I do not necessarily agree with the authors suggestion about a 90 mm threshold, but they have defended their position and I am satisfied by the way they caveat their results and interpretations. Personally I think it is unlikely that many dispersals were happening when rainfall was below 200 or more mm – but the point is that their data and figures etc can be used in discussing such a position. In other words, there is a clear distinction between the data and their interpretation of it. This is how it should be with all papers, but all too often in such topics there is no separation of description and interpretation.

One very minor thing that I would suggest is that given the authors argue for an “an unprecedentedly long period of largely favourable climate between ~65k and ~30k years ago” in southern Arabia, they should mention that despite decades of study, no evidence of this has been found. There is evidence for minor wet phases such as ca 55 ka, but not for a ca. 35 thousand year wet phase that is a major conclusion of their study. This is not to say that this wet phase did not happen – but climate archives showing it have not been found. As suggested in my previous review, I think the way they have calibrated their climate model against possibly inappropriate archives (e.g. dubious to understand monsoonal dynamics using the Dead Sea record) may be the reason for an anomalously prolonged southern Arabian wet phase ca. 65 to 35 ka. I would appreciate if they mentioned an absence of evidence for a long southern Arabian humid period in the proposed time. They could mention that testing such a model should be a priority for future palaeoclimate research in the region. An alternative interpretation of the major dispersal ca. 70-50 ka is that a small population which had passed through the arid belt expanded, spread and split; they may have already been in more humid regions. So the apparent correlation between the ‘successful’ dispersal period and the hypothesized 65-30 ka major southern Arabian wet phase is tempting, but I would appreciate if some caveats with briefly mentioned.

We have added the following paragraph to the text to accommodate the Reviewer’s comment:

We note that this latter scenario has been a subject of debate based on the empirical palaeoenvironmental record ⁴², with conclusions ranging from the Arabian Peninsula being continually too arid for human migration ²¹, to intermittent wet intervals ^{43,44}, and extended pluvial periods ⁴⁵ during marine isotope stage 3 (57–29k years ago). In any case, these inferences are not directly comparable with our results, both because several empirical proxies are not suitable for detecting rainfall of the small magnitude considered here (e.g. speleothems ⁴⁶), and because, for each route, the specific path out of Africa that requires the least tolerance to low precipitation out of all possible paths varies over time, as does the geographic location of its driest segment, whose rainfall level is shown in Fig. 1; thus, our estimates would not be expected to necessarily display the same patterns over time as a localised empirical climate reconstruction.

Minor points.

‘.’ missing in email address for corresponding author on main text file.

We have corrected the email address.

They say Arabia was 'likely visible' from Africa. It absolutely would have been visible.

We removed "likely".

"competing with Neanderthal" should be Neanderthals.

We have corrected the typo.

"Denisova" should be Denisovans.

We have changed the spelling, as suggested.

I don't understand what the 'Nefud' lake is in supplementary figure 4. And no reference is provided. There are many lakes in the Nefud, of lots of different ages.

We have replaced the lake names in the figure by number identifiers and now describe the lakes in the text in terms of their location ("western Nefud near Taymal", "at Khujaymah", and "at Saiwan") following the description in the referenced paper of Jennings, R. P. et al. (2015).

In summary, I have these few minor suggestions for edits, otherwise I think the paper is ready and should be published.